# *Few-Class Arena*: A Benchmark for Efficient Selection of Vision Models and Dataset Difficulty Measurement

**Bryan Bo Cao**[*] **& Shubham Jain**[†]
Department of Computer Science
Stony Brook University
Stony Brook, NY, USA
`{boccao,jain}@cs.stonybrook.edu`

**Lawrence O'Gorman**[†] **& Michael Coss**
Nokia Bell Labs
Murray Hill, NJ, USA
`{larry.o_gorman,mike.coss}@nokia-bell-labs.com`

## ABSTRACT

We propose *Few-Class Arena* (*FCA*), as a unified benchmark with focus on testing efficient image classification models for few classes. A wide variety of benchmark datasets with many classes (80-1000) have been created to assist Computer Vision architectural evolution. An increasing number of vision models are evaluated with these many-class datasets. However, real-world applications often involve substantially fewer classes of interest (2-10). This gap between many and few classes makes it difficult to predict performance of the few-class applications using models trained on the available many-class datasets. To date, little has been offered to evaluate models in this *Few-Class Regime*. We conduct a systematic evaluation of the ResNet family trained on ImageNet subsets from 2 to 1000 classes, and test a wide spectrum of Convolutional Neural Networks and Transformer architectures over ten datasets by using our newly proposed *FCA* tool. Furthermore, to aid an up-front assessment of dataset difficulty and a more efficient selection of models, we incorporate a difficulty measure as a function of class similarity. *FCA* offers a new tool for efficient machine learning in the *Few-Class Regime*, with goals ranging from a new efficient class similarity proposal, to lightweight model architecture design, to a new scaling law. *FCA* is user-friendly and can be easily extended to new models and datasets, facilitating future research work. Our benchmark is available at https://github.com/bryanbocao/fca.

## 1 INTRODUCTION

The de-facto benchmarks for evaluating efficient vision models are large scale with many classes (e.g. 1000 in ImageNet (Deng et al., 2009), 80 in COCO (Lin et al., 2014), etc.). Such benchmarks have expedited the advance of vision neural networks toward efficiency (Tan & Le, 2019a; Tan & L., 2021; Sinha & El-Sharkawy, 2019; Sandler et al., 2018; Howard et al., 2019; Iandola et al., 2016; Ma et al., 2018; Mehta & Rastegari) with the hope of reducing the financial and environmental cost of vision models (Patterson et al., 2021; Rae et al., 2021). More efficient computation is facilitated by using quantization (Gysel et al., 2018; Han et al., 2015; Leng et al., 2018), pruning (Cheng et al., 2017; Blalock et al., 2020; Li et al., 2016; Shen et al., 2022), and data saliency (Yeung et al., 2016). Despite efficiency improvements such as these, many-class datasets are still the standard of model evaluation.

---

[*]Work done during an internship at Nokia Bell Labs.
[†]Corresponding authors.

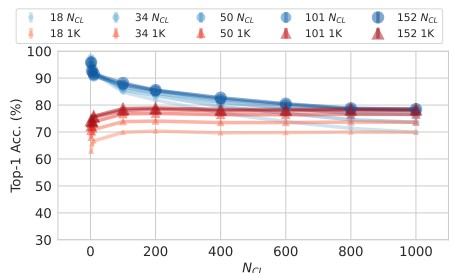

(a) Accuracies for sub-models (blue) and full models (red).

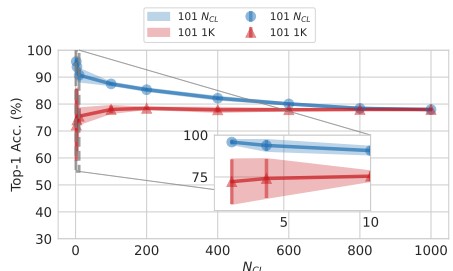

(b) Zoomed window shows accuracy values and range for full and sub-models in the few-class range.

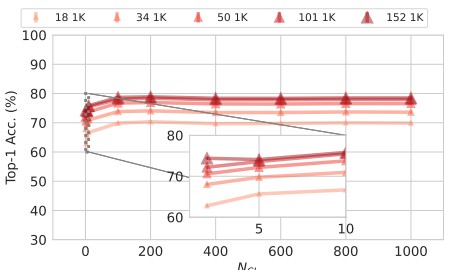

(c) Zoomed window shows (c.1) drop of accuracy as $N_{CL}$ decreases, (c.2) accuracy scales with model size for full models in the few-class range.

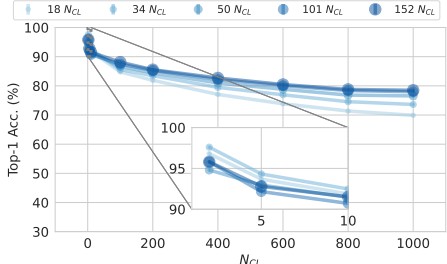

(d) Zoomed window shows (d.1) rising accuracy as $N_{CL}$ decreases, (d.2) accuracy does not scale with model size for sub-models in the few-class range.

Figure 1: Top-1 accuracies of various scales of ResNet, whose model sizes are shown in the legend, and whose plots vary from dark to light by decreasing size. Plots range along number of classes $N_{CL}$ from the full ImageNet size (1000) down to the *Few-Class Regime*. Each model is tested on 5 subsets whose $N_{CL}$ classes are randomly sampled from the original 1000 classes. (a) Plots for sub-models trained on subsets of classes (blue) and full models trained on all 1000 classes (red). (b) Zoomed window shows the standard deviation of subset's accuracies is much smaller than for the full model. (c.1) Full model accuracies drop when $N_{CL}$ decreases. (c.2) Full model accuracies increase as model scales up in the *Few-Class Regime*. (d.1) Sub-model accuracies grow as $N_{CL}$ decreases. (d.2) Sub-model accuracies do not increase when model scales up in the *Few-Class Regime*.

Real-world applications, however, typically comprise only a few number of classes (e.g, less than 10) (Shao et al., 2020; A. Delplanque, 2022; Cai et al., 2021) which we termed *Few-Class Regime*. This introduces a crucial research question: what is the simplest baseline model capable of meeting performance criteria within this *Few-Class Regime*? To deploy a vision model pre-trained on large datasets in a specific environment, it requires the re-evaluation of published models or even retraining to find an optimal model in an expensive architectural search space (Scheidegger et al., 2019).

One major finding is that, apart from scaling down model and architectural design for efficiency, dataset difficulty also plays a vital role in model selection (Scheidegger et al., 2021) (described in Section 4.3).

Figure 1 summarizes several key findings under the *Few-Class Regime*. On the bottom left graph in red are accuracy results for a range of number of classes $N_{CL}$ for what we call the "full model", that is ResNet models pre-trained on the full 1000 classes of ImageNet (generally available from many websites). On the bottom right in blue are accuracy results for what we call "sub-models", each of which is trained and tested on the same $N_{CL}$, where this number of classes is sampled from the full dataset down to the *Few-Class Regime*. Findings include the following. (a) Sub-models attain higher upper-bound accuracy than full models. (b) The range of accuracy widens for full models at few-classes, which increases the uncertainty of a practitioner selecting a model for few classes. In contrast, sub-models narrow the range. (c) Full models follow the scaling law (Kaplan

et al., 2020) in the dimension of model size - larger models (darker red) have higher accuracy from many to few classes. (d) Surprisingly, the scaling law is violated for sub-models in the *Few-Class Regime* (see the zoomed-in subplot) where larger models (darker blue) do not necessarily perform better than smaller ones (lighter blue). From these plots, our key insight is that, instead of using full models, researchers and practitioners in the *Few-Class Regime* should use sub-models for selection of more efficient models.

However, obtaining sub-models involves computationally expensive training and testing cycles since they need to be converged on each of the few-class subsets. By carefully studying and comparing the experiment and evaluation setup of these works in the literature, we observe that, how models scale down to *Few-Class Regime* is rarely studied. The lack of comprehensive benchmarks for *few-class* research impedes both researchers and practitioners from quickly finding models that are the most efficient for their dataset size. To fill this need, we propose a new benchmark, *Few-Class Arena* (*FCA*), with the goal of benchmarking vision models under few-class scenarios. To our best knowledge, *FCA* is the first benchmark for such a purpose.

We formally define *Few-Class Regime* as a scenario where the dataset has a limited number of classes. Real-world applications often comprise only a few number of classes (e.g. $N_{CL} < 10$ or $10\%$ classes of a dataset). Consequently, *Few-Class Arena* refers to a benchmark to conduct research experiments to compare models in the *Few-Class Regime*. This paper focuses on the image classification task, although *Few-Class Regime* can generalize to object detection (Chen et al., 2019), segmentation (Contributors, 2020) and other visual tasks (described in the Appendix).

**Statement of Contributions.** Four contributions are listed below:

- To the best of our knowledge, we are the first to explore the problems in the *Few-Class Regime* and develop a benchmark tool *Few-Class Arena* (*FCA*) to facilitate scientific research, analysis, and discovery for this range of classes.
- We introduce a scalable few-class data loading approach to automatically load images and labels in the *Few-Class Regime* from the full dataset, avoiding the need to duplicate data points for every additional few-class subset.
- We incorporate dataset similarity as an inverse difficulty measurement in *Few-Class Arena* and propose a novel Silhouette-Based Similarity Score named *SimSS*. By leveraging the visual feature extraction power of CLIP and DINOv2, we show that *SimSS* is highly correlated with ResNet performance in the *Few-Class Regime* with high Pearson coefficient scores $\geq 0.88$.
- We conduct extensive experiments that comprise ten models on ten datasets and 2-1000 numbers of classes on ImageNet, totalling 1591 training and testing runs. In-depth analyses on this large body of testing reveal new insights in the *Few-Class Regime*.

## 2 RELATED WORK

**Visual Datasets and Benchmarks.** To advance deep neural network research, a wealth of large-scale many-class datasets has been developed for benchmarking visual neural networks over a variety of tasks. Typical examples [1] include 1000 classes in ImageNet (Deng et al., 2009) for image classification, and 80 object categories in COCO (Lin et al., 2014) for object detection. Previous benchmarks also extend vision to multimodal research such as image-text (Lee et al., 2024; Le et al., 2024; Laurençon et al., 2024; Bitton et al., 2022). While prior works often scale up the number of object categories for general purpose comparison, studies (Fang et al., 2024; Mayo et al., 2023) raise a concern on whether models trained on datasets with such a large number of classes (e.g. ImageNet) can be reliably transferred to real world applications often with far fewer classes. A close work to ours is vision backbone comparison (Goldblum et al., 2024) whose focus is on model architectures. Our perspective differs in a focus on cases with fewer number of classes, which often better aligns with real-world scenarios.

**Dataset Difficulty Measurement.** Research has shown the existence of inherent dataset difficulty (Mayo et al., 2023) for classification and other analytic tasks. Efficient measurement methods are proposed to characterize dataset difficulty using Silhouette Score (Rousseeuw, 1987), K-means

---

[1] A detailed list of many-class datasets used in this paper can be found in the Appendix.

Fréchet inception distance (Dowson & Landau, 1982; Heusel et al., 2017; Lucic et al., 2018), and Probe nets (Scheidegger et al., 2021). Prior studies have proposed image quality metrics using statistical heuristics, including peak signal-to-noise ratio (PSNR) (Hore & Ziou, 2010), structural similarity (SSIM) Index (Wang et al., 2004), and visual information fidelity VIF (Sheikh & Bovik, 2006). A neuroscience-based image difficulty metric (Mayo et al., 2023) is defined as the minimum viewing time related to object solution time (OST) (Kar et al., 2019). Another type of difficulty measure method consists of additional procedures such as c-score (Jiang et al., 2020), prediction depth (Baldock et al., 2021), and adversarial robustness (Goodfellow et al., 2014). Our work aligns with the line of research (Arun, 2012; Trick & Enns, 1998; Wolfe et al., 2010) involving similarity-based difficulty measurements: similar images are harder to distinguish from each other while dissimilar images are easier. Previous studies are mainly in the image retrieval context (Zhang & Lu, 2003; Wang et al., 2014; Tudor Ionescu et al., 2016). Similarity score is used in (Cao et al., 2023) with the limitation that a model serving similarity measurement has to be trained for one dataset. We push beyond this limit by leveraging large vision models that learn general visual features using CLIP (Radford et al., 2021) and DINOv2 (Oquab et al., 2023). The study (Mayo et al., 2023) shows that CLIP generalizes well to both easy and hard images, making it a good candidate for measuring image difficulty. Supported by the evidence that better classifiers can act as better perceptual feature extractors (Kumar et al., 2022), in later sections we show how CLIP and DINOv2 will be used as our similarity base function.

Despite the innovation of difficulty measure algorithms on many-class datasets, little attention has been paid to leveraging these methods in the *Few-Class Regime*. We show that, as the number of classes decreases, sub-dataset difficulty in the *Few-Class Regime* plays an increasingly critical role in efficient model selection. To summarize, unlike previous work on many-class benchmarks and difficulty measurements, our work takes few-class and similarity-based dataset difficulty into consideration, and in doing so we believe the work pioneers the development of visual benchmark dedicated to research in the *Few-Class Regime*.

## 3 FEW-CLASS ARENA (FCA)

We introduce the *Few-Class Arena* (*FCA*) benchmark in this section. In practice, we have integrated *FCA* into the MMPreTrain framework (Contributors, 2023), implemented in Python and Pytorch[2]. Our benchmark usage guidelines are detailed in A.2 of the Appendix.

### 3.1 FEW-CLASS DATASET PREPARATION

*Few-Class Arena* provides an easy way to prepare datasets in the *Few-Class Regime*. By leveraging the MMPreTrain framework, users only need to specify the parameters of few-class subsets in the configuration files, which includes the list of models, datasets, number of classes ($N_{CL}$), and the number of seeds ($N_S$). *Few-Class Arena* generates the specific model and dataset configuration files for each subset, where subset classes are randomly extracted from the full set of classes, as specified by the seed number. Note that only one copy of the full, original dataset is maintained during the whole benchmarking life cycle because few-class subsets are created through the lightweight configurations, thus maximizing storage efficiency. We refer readers to the Appendix and the publicly released link for detailed implementations and use instructions.

### 3.2 MANY-CLASS FULL DATASET TRAINED BENCHMARK

We conducted large-scale experiments spanning ten popular vision models (including CNN and ViT architectures) and ten common datasets [3]. Except for ImageNet1K, where pre-trained model weights

---

[2]Code is available at https://github.com/bryanbocao/fca, including detailed documentation and long-term plans of maintenance.

[3]Models include: ResNet50 (RN50), VGG16, ConvNeXt V2 (CNv2), Inception V3 (INCv3), EfficientNet V2 (EFv2), ShuffleNet V2 (SNv2), MobileNet V3 (MNv3), Vision Transformer base (ViTb), Swin Transformer V2 base (SWv2b) and MobileViT small (MViTs). Datasets include CalTech101 (CT101), CalTech256 (CT256), CIFAR100 (CF100), CUB200 (CB200), Food101 (FD101), GTSRB43 (GT43), ImageNet1K (IN1K), Indoor67 (ID67), Quickdraw345 (QD345) and Textures47 (TT47).

are available, we train models in other datasets from scratch. While different models' training procedures may incur various levels of complexity (particularly in our case for MobileNet V3 and Swin Transformer V2 base), we have endeavored to minimize changes in the existing training pipelines from MMPreTrain. The rationale is that if a model exhibits challenges in adapting it to a dataset, then it is often not a helpful choice for a practitioner to select for deployment.

Results are summarized in Table 1. We observe (1) models in different datasets (in rows) yield highly variable levels of performance by Top-1 accuracy; (2) no single best model (bold, in columns) exists across all datasets; and (3) model rankings vary across various datasets.

The first two observations are consistent with the findings in (Scheidegger et al., 2021; Fang et al., 2024). For (1), it suggests there exists underlying dataset-specific difficulty. To capture this characteristic, we adopt the reference dataset classification difficulty number (DCN) (Scheidegger et al., 2021) to refer to the empirically highest accuracy achieved in a dataset from a finite number of models shown in Table 1 and Figure 2 (a). For observation (3), we can examine the rankings among the ten datasets of ResNet50 and EfficientNet V2 in Figure 2 (b). ResNet50's ranking varies dramatically across different datasets, for instance ranking 7th on ImageNet1K and 1st on Quickdraw345. This ranking variability is also observed in other models (see all models in the Appendix). However, a common practice is to benchmark models – even for efficiency – on large datasets, especially ImageNet1K. The varied dataset rankings in our experiments expose the limitations of such a practice, further supporting our new benchmark paradigm, especially in the *Few-Class Regime*. In later sections, we leverage DCN and image similarity for further analysis.

| Dataset | RN50 | VGG16 | CNv2 | INCv3 | EFv2 | SNv2 | MNv3 | ViTb | SWv2b | MViTs | DCN |
|---|---|---|---|---|---|---|---|---|---|---|---|
| GT43 | 99.85 | 96.60 | 99.83 | 99.78 | 99.86 | **99.87** | 99.83 | 99.31 | 99.78 | 99.69 | 99.87 |
| CF100 | 74.56 | 71.12 | **85.89** | 75.97 | 77.05 | 77.89 | 74.35 | 32.65 | 78.49 | 76.51 | 85.89 |
| IN1K | 76.55 | 71.62 | 84.87 | 77.57 | **85.01** | 69.55 | 66.68 | 82.37 | 84.60 | 78.25 | 85.01 |
| FD101 | 83.76 | 75.82 | 63.80 | 83.96 | 80.82 | 79.36 | 76.03 | 52.21 | **84.30** | 82.23 | 84.30 |
| CT101 | 77.70 | 74.99 | 77.52 | 77.52 | 77.82 | **84.13** | 80.71 | 59.59 | 78.82 | 80.06 | 84.13 |
| CT256 | 65.07 | 59.08 | **73.57** | 66.09 | 62.80 | 68.13 | 62.62 | 44.23 | 67.28 | 65.80 | 73.57 |
| QD345 | **69.14** | 19.86 | 62.86 | 68.25 | 68.81 | 67.32 | 66.42 | 19.67 | 66.54 | 68.76 | 69.14 |
| CB200 | 45.86 | 21.26 | 27.61 | 45.58 | 44.48 | 53.95 | 53.80 | 23.73 | 54.52 | **58.46** | 58.46 |
| ID67 | 53.75 | 26.01 | 33.21 | 45.95 | 43.85 | **54.72** | 51.65 | 30.51 | 48.58 | 54.05 | 54.72 |
| TT47 | 30.43 | 12.55 | 6.49 | 14.20 | 21.17 | **43.83** | 40.27 | 31.38 | 33.94 | 24.41 | 43.83 |

Table 1: Top-1 accuracy across ten models in ten datasets. Models are trained and tested on full datasets with their original number of classes (e.g. 1K from ImageNet1K) denoted in the last few digits of the abbreviation of the dataset name. The best score is highlighted in bold while the second best is underlined for each dataset. References for all models and datasets are in the Appendix.

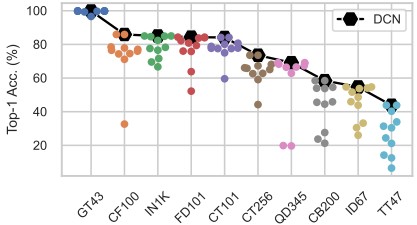

(a) Top-1 accuracy and DCN in ten full datasets.

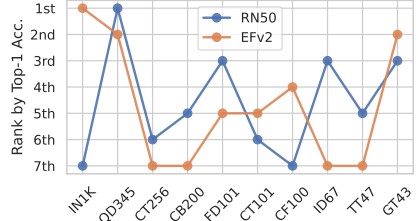

(b) Ranking of ResNet50 (RN50) and EfficientNet V2 (EFv2) across 10 datasets by Top-1 acc.

Figure 2: Many-Class Full Dataset Benchmark.

In the next subsections, we introduce three new types of benchmarks: (1) Few-Class Full Dataset Trained Benchmark (FC-Full), which benchmarks vision models trained on the full dataset with the original number of classes; (2) Few-Class Subset Trained Benchmark (FC-Sub), which benchmarks vision models trained on subsets of a fewer number of classes than the full dataset, and (3) Few-Class

Similarity Benchmark (FC-Sim), which benchmarks image similarity methods and their correlation with model performance.

## 3.3 Few-Class Full Dataset Trained Benchmark (FC-Full)

Traditionally, a large number of models are trained and compared on many-class datasets. However, such results cannot be directly transferred to the *Few-Class Regime* and many real-world scenarios. Therefore, we introduce the Few-Class Full Dataset Trained Benchmark (FC-Full), with the objective of effortlessly conducting large-scale experiments and analyses in the *Few-Class Regime*.

The procedure of FC-Full consists of two main stages. In the first stage, users select the models and datasets upon which they would like to conduct experiments. They can choose to download pre-trained model weights, which are usually available on popular model hubs (PyTorch Hub (Foundation, 2024), TensorFlow Hub (Inc., 2024), Hugging Face (Face, 2024), MMPreTrain (Contributors, 2023) etc.). In case of no pre-trained weights available from public websites, users can resort to the option of training from scratch. To that end, our tool is designed and implemented to generate bash scripts for easily configurable and modifiable training through the use of configuration files.

In the second stage, users conduct benchmarking in the *Few-Class Regime*. By specifying the list of classes, *Few-Class Arena* automatically loads pre-trained weights of the chosen models and evaluates performance of the models on the selected datasets. Note that this process is accomplished through configuration files created by the user's specifications, thus enabling hundreds of experiments to be launched by a single command. This dramatically reduces the human effort that would otherwise be expended to run these experiments without *Few-Class Arena*.

## 3.4 Few-Class Subset Trained Benchmark (FC-Sub)

Our study in Figure 1 (red lines) reveals the limits of existing pre-trained models in the *Few-Class Regime*. To facilitate further research and analyze the upper bound performance in the *Few-Class Regime*, we introduce the Few-Class Subset Trained Benchmark (FC-Sub).

FC-Sub follows a similar procedure to FC-Full, except that, when evaluating a model in a subset with a specific number of classes, that model should have been trained on that same subset. Specifically, in Stage One (described for FC-Full), users specify models, datasets and the list of number of classes in configuration files. Then *Few-Class Arena* generates bash scripts for model training on each subset. In Stage two, *Few-Class Arena* tests each model in the same subset that it was trained on.

## 3.5 Few-Class Similarity Benchmark (FC-Sim)

One objective of our tool is to provide the Similarity Benchmark as a platform for researchers to design custom similarity scores for efficient comparison of models and datasets.

The intrinsic image difficulty of a dataset affects a model's classification performance (and human) (Geirhos et al., 2017; Rajalingham et al., 2018; Mayo et al., 2023). We show – as is intuitive – that the more similar two images are, the more difficult it is for a vision classifier to make a correct prediction. This suggests that the level of similarity of images in a dataset can be used as a proxy for a dataset difficulty measure. In this section, we first adopt and provide the basic formulation of similarity, the baseline of a similarity metric. Then we propose a Similarity-Based Silhouette Score to capture the characteristic of image similarity in a dataset.

We first adopt the basic similarity formulation from (Cao et al., 2023). **Intra-Class Similarity** $S_\alpha^{(C)}$ is defined as a scalar describing the similarity of images within a class by taking the average of all the distinct class pairs in $C$, while **Inter-Class Similarity** denotes a scalar describing the similarity among images in two different classes $C_1$ and $C_2$. For a dataset $D$, these are defined as the mean of their similarity scores over all classes, respectively:

$$S_\alpha^{(D)} = \frac{1}{|L|} \sum_{l \in L} S_\alpha^{(C_l)} = \frac{1}{|L| \times |P^{(C_l)}|} \sum_{l \in L} \sum_{i,j \in C_l; \, i \neq j} \cos(\mathbf{Z}_i, \mathbf{Z}_j), \qquad (1)$$

$$S_\beta^{(D)} = \frac{1}{|P^{(D)}|} \sum_{a,b \in L; a \neq b} S_\beta^{(C_a, C_b)} = \frac{1}{|P^{(D)}| \times |P^{(C_1,C_2)}|} \sum_{a,b \in L; \, a \neq b} \sum_{i \in C_1, j \in C_2} \cos(\mathbf{Z}_i, \mathbf{Z}_j), \quad (2)$$

where $|L|$ is the number of classes in a dataset, $Z_i$ is the visual feature of an image $i$, $|P^{(C)}|$ is the total number of distinct image pairs in class $C$, $|P^{(D)}|$ is the total number of distinct class pairs, and $|P^{(C_1,C_2)}|$ is the total number of distinct image pairs excluding same-class pairs.

Averaging these similarities provides a single scalar score at the class or dataset level. However, this simplicity neglects other cluster-related information that can better reveal the underlying dataset difficulty property of a dataset. In particular, the **(1) tightness of a class cluster** and **(2) distance to other classes** of class clusters, are features that characterize the inherent class difficulty, but are not captured by $S_\alpha$ or $S_\beta$ alone.

To compensate the aforementioned drawback, we adopt the Silhouette Score (SS) (Rousseeuw, 1987; Shahapure & Nicholas, 2020): $SS(i) = \frac{b(i)-a(i)}{max(a(i),b(i))}$, where $SS(i)$ is the Silhouette Score of the data point $i$, $a(i)$ is the average dissimilarity between $i$ and other instances in the same class, and $b(i)$ is the average dissimilarity between $i$ and other data points in the closest different class.

Observe that the above Intra-Class Similarity $S_\alpha^{(C)}$ already represents the tightness of the class $(C)$, therefore $a(i)$ can be replaced with the inverse of Intra-Class Similarity $a(i) = -S_\alpha(i)$. For the second term $b(i)$, we adopt the previously defined Inter-Class Similarity $S_\beta^{(C_1,C_2)}$ and introduce a new similarity score as **Nearest Inter-Class Similarity** $S'_\beta{}^{(C)}$, which is a scalar describing the similarity among instances between class $C$ and the closest class of each instance in $C$. The dataset-level Nearest Inter-Class Similarity $S'_\beta{}^{(D)}$ is expressed as:

$$S'_\beta{}^{(D)} = \frac{1}{|L|} \sum_{l \in L} S'_\beta{}^{(C_l)} = \frac{1}{|L| \times |P^{(C_l,\hat{C}_l)}|} \sum_{l \in L} \sum_{i \in C_l, j \in \hat{C}_l} \cos(\mathbf{Z}_i, \mathbf{Z}_j), \tag{3}$$

where $\hat{C}$ is the nearest class to instance $i$ ($\hat{C} \neq C$). To summarize, we introduce our novel **Similarity-Based Silhouette Score** $SimSS$[4] for dataset $D$:

$$SimSS^{(D)} = \frac{1}{|L| \times |C_l|} \sum_{i \in C_l} \frac{S_\alpha(i) - S'_\beta(i)}{max(S_\alpha(i), S'_\beta(i))}. \tag{4}$$

# 4 EXPERIMENTAL RESULTS

## 4.1 RESULTS ON FC-FULL

In this section, we present the results of FC-Full. A model trained on the dataset with its original number of classes (e.g. 1000 in ImageNet1K) is referred to as a *full-class model*. These experiments are designed to understand how full-class model performance changes when the number of classes $N_{CL}$ decreases from many to few classes. We analyze the results of DCN-Full, shown in Figure 3 (details of all models are presented in the Appendix), and we make two key observations when $N_{CL}$ reduces to the *Few-Class Regime* (from right to left). (1) The best performing models do not always increase its accuracy for fewer classes, as shown by the solid red lines that represent the average of DCN for each $N_{CL}$. (2) The variance, depicted by the light red areas, of the best models broaden dramatically for low $N_{CL}$, especially for $N_{CL} < 10$.

Both observations support evidence of the limitations of using the common many-class benchmark for application model selection in the *Few-Class Regime*, since it is not consistent between datasets that a model can be made smaller with higher accuracy. Furthermore, the large variance in accuracy means that prediction of performance for few classes is unreliable for this approach.

## 4.2 RESULTS ON FC-SUB

In this section, we show how using *Few-Class Arena* can help reveal more insights in the *Few-Class Regime* to mitigate the issues of Section 4.1.

FC-Sub results are displayed in Figure 4. Recall that a *sub-class* model is a model trained on a subset of the dataset where $N_{CL}$ is smaller than the original number of classes in the full dataset.

---

[4]The extended derivation is detailed in the Appendix.

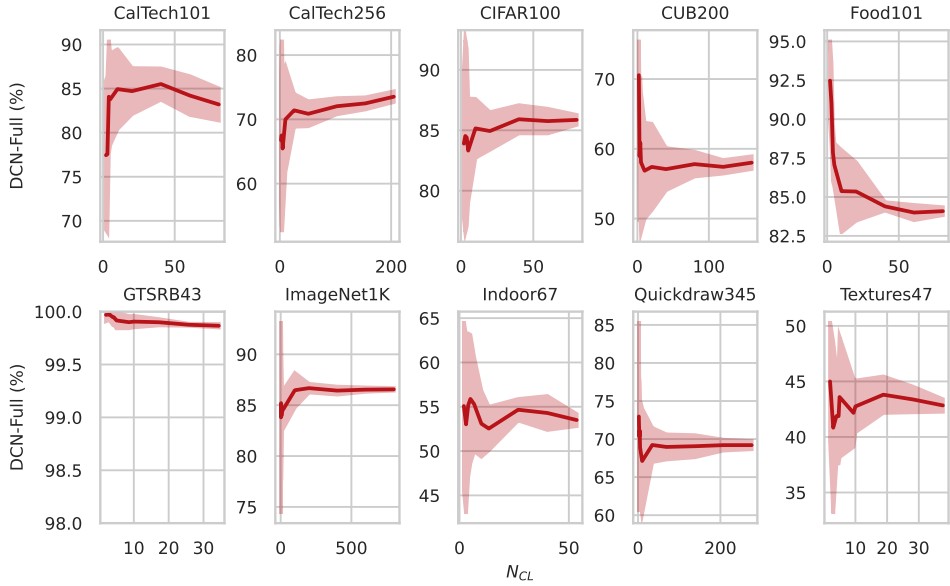

Figure 3: DCN-Full by Top-1 Accuracy (%). $N_{CL}$ ranges from many to 2.

Observe that in the *Few-Class Regime* (when $N_{CL}$ decreases from 4 to 2) that: (1) DCN increases as shown by the solid blue lines, and (2) variance reduces as displayed by the light blue areas.

The preceding observation for FC-Full 4.1 seems to contradict the common belief that, the fewer the classes, the higher the accuracy a model can achieve. Conversely, the FC-Sub results do align with this belief. We argue that a full-class model needs to accommodate many parameters to learn features that will enable high performance across all classes in a many-class, full dataset. With the same parameters, however, a sub-class model can adapt to finer and more discriminative features that improve its performance when the number of target classes are much smaller.

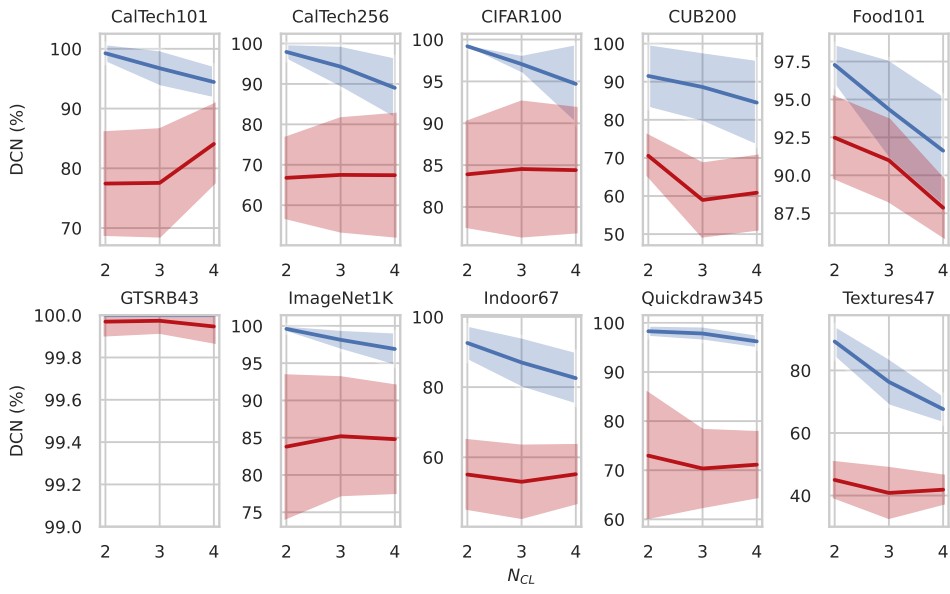

Figure 4: DCN-Sub (blue) and DCN-Full (red) by Top-1 Accuracy (%). $N_{CL}$ ranges from 2 to 4.

### 4.3 RESULTS ON FC-SIM

In this section, we analyze the use of SimSS (Equation 4) as proxy for few-class dataset difficulty. Experiments are conducted on ImageNet1K using the ResNet family for the lower $N_{CL} \leq 10\%$ range of the original 1000 classes, $N_{CL} \in \{2, 3, 4, 5, 10, 100\}$, and the results are shown in Figure 5. Each datapoint of DCN-Full (diamond in red) or DCN-Sub (square in blue) represents an experiment in a subset of a specific $N_{CL}$, where classes are sampled from the full dataset. For reproducible results, we use seed numbers from 0 to 4 to generate 5 subsets for one $N_{CL}$ by default. A similarity base function ($sim()$) is defined as the atomic function that takes a pair of images as input and outputs a scalar that represents their image similarity.

In our experiments, we leverage the general visual feature extraction ability of CLIP (image + text) (Radford et al., 2021) and DINOv2 (image) (Oquab et al., 2023) by self-supervised learning. Specifically, a pair of images are fed into its latent space from which the the cosine score is calculated and normalized to 0 to 1. Note that we only use the Image Encoder in CLIP.

**Comparing Accuracy and Similarity** To evaluate SimSS, we compute the Pearson correlation coefficient (PCC) ($r$) between model accuracy and SimSS. Results in Figure 5 (a) (b) show that SimSS is poorly correlated with DCN-Full ($r = 0.18$ and $r = 0.26$ for CLIP and DINOv2) due to the large variance shown in Section 4.1 , as well as the general features learned on a full dataset, those of which can be extraneous to a much smaller set of target sub-class features. In contrast, SimSS is highly correlated with DCN-Sub (shown in blue squares), with $r = 0.90$ and $r = 0.88$ using CLIP (dashed) and DINOv2 (solid), respectively. We attribute the advantages of DCN-Sub to its focus on the minimal features tailored to the target sub-classes, while maintaining the same number of parameters as the DCN-Full architecture. The high PCC (Wicklin, 2024; Schober et al., 2018) demonstrates that SimSS is a reliable metric to estimate few-class dataset difficulty, and this can help predict the empirical upper-bound accuracy of a model in the *Few-Class Regime*. Comparison between SimSS and all models can be found in the Appendix. Such a high correlation suggests this offers a reliable scaling relationship to estimate model accuracy by similarity for other values of $N_{CL}$ without an exhaustive search. Due to the dataset specificity of the dataset difficulty property, this score is computed once and used for all times the same dataset is used. We have made available difficulty scores for many datasets at the *Few-Class Arena* site.

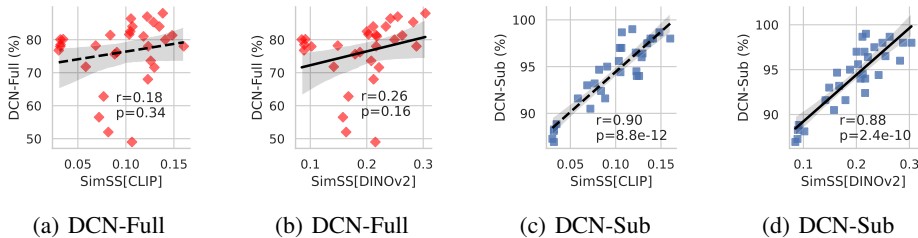

| (a) DCN-Full | (b) DCN-Full | (c) DCN-Sub | (d) DCN-Sub |

Figure 5: Pearson correlation coefficient ($r$) between DCN and SimSS when $N_{CL} \in \{2, 3, 4, 5, 10, 100\}$. DCN-Sub (blue squares) is more highly correlated than DCN-Full (red diamonds) with SimSS using both similarity base functions of CLIP (dashed line) and DINOv2 (solid line) with $r \geq 0.88$.

### 4.4 COMPARISON WITH FINE-TUNED MODELS

Fine-tuning a model through transfer learning from a pre-trained model has become as a common practice in many real-world scenarios. We perform experiments on CNNs (ResNet18, 50) and Transformer architectures (MobileVit-small (Mehta & Rastegari, 2021), ViT-Base) summarized in Table 2. To fine-tune (FT) the ResNet18, 50 and MobileVit-small models for $N_{CL} \in \{2, 4\}$, we first train their full models $N_{CL} = \{100\}$ on CIFAR100 for 100 epochs, and subsequently fine-tune their weights for the target $N_{CL} \in \{2, 4\}$ for another 20 epochs. For ViT, we fine-tune a ViT-B model initialized with weights from the CLIP pre-trained backbone. A linear layer is added on top, and the model is trained for 10 epochs. This setup is indicated by the star symbol (*). We conclude that the fine-tuned models exhibit patterns and trends consistent with the observations presented in Fig. 1. Note our focus of this work is to leverage the proposed difficult measurement method, FC-Sim, to

efficiently estimate the achievable model accuracy, thereby assisting in model selection in the *Few-Class Regime*. Sub-models can offer insights into the minimal visual features required for a specific real-world scenario as they are trained exclusively on the target classes. In contrast, weights pre-trained on large full datasets – whether through fully supervised manner or self-supervised – may include extraneous features that are irrelevant to the target classes. We therefore prioritize sub-model study in this work.

| MT | $N_{CL}$ | ResNet18 | ResNet50 | MViT-S | ViT-B |
|----|------|----------|----------|--------|-------|
| F | 100 | 76.11 | 73.71 | 73.83 | 32.54 |
| F | 4 | 75.10 | 72.20 | 72.35 | 36.15 |
| FT | 4 | 87.60 | **90.55** | **90.00** | **91.16**\* |
| S | 4 | **90.65** | 90.15 | 89.45 | 85.40 |
| F | 2 | 75.00 | 71.30 | 71.80 | 40.80 |
| FT | 2 | 87.90 | 93.70 | 90.50 | 95.20\* |
| S | 2 | **96.30** | **95.30** | **95.50** | **95.90** |

Table 2: Top-1 Accuracies for different configurations on CIFAR100. $N_{CL} \in \{2, 4, 100\}$, MT: Model Type, F: Full model, S: Sub-model, FT: Fine-tuned model, MViT-S: MobileViT-small, \*: fine-tuned from the CLIP pre-trained model. Best scores are highlighted in bold. The gray bar indicates sub-models as the primary focus of this research.

## 5 CONCLUSION

We have proposed *Few-Class Arena* and a dataset difficulty measurement, which together form a benchmark tool to compare and select efficient models in the *Few-Class Regime*. Extensive experiments and analyses over 1500 experiments with ten models on ten datasets have helped identify new behaviors specific to the *Few-Class Regime* as compared to many-classes setting. One finding reveals a new $N_{CL}$-scaling law whereby dataset difficulty must be taken into consideration for accuracy prediction. Such a benchmark will be valuable to the community by providing both researchers and practitioners with a unified framework for future research and real applications.

**Limitations and Future Work.** The current difficulty benchmark supports image similarity while in the future it can be expanded to other difficulty measurements (Scheidegger et al., 2021). CLIP and DINOv2 are trained toward general visual features, it is unclear if they will be appropriate for other types of images such as sketches without textures in Quickdraw (Ha & Eck, 2017). For this reason, a universal similarity foundation model would be appealing that applies to any image type. Moreover, integrating image similarity with representation similarity (AntixK, 2023; Nguyen et al., 2020; Cao et al., 2024) could further enhance model efficiency, leveraging complementary insights from both approaches. In summary, *Few-Class Arena* identifies a promising new path for achieving efficiencies focused on the important and practical *Few-Class Regime*, establishing a baseline for future work.

## ACKNOWLEDGMENTS

This research has been supported in part by the National Science Foundation (NSF) under Award numbers 2055520 and 2238553. We appreciate the GPU-cluster maintenance support from Rob Dinoff and Hans Woithe for running large-scale experiments, as well as insightful discussions with Zhengyu Wu and Xi Han significantly enhanced the clarity and readability of our plots.

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

# A  APPENDIX

## A.1  GOALS

**1. Generality.** All vision models and existing datasets for classification should be compatible in this framework. In addition, users can extend to custom models and datasets for their needs.

**2. Efficiency.** The benchmark should be time- and space-efficient for users. The experimental setup for the few-class benchmark should be easily specified by a few hyper-parameters (e.g. number of classes). Since the few-class regime usually includes sub-datasets extracted from the full dataset, the benchmark should be able to locate those sub-datasets without generating redundant duplicates for reasons of storage efficiency. For time-efficiency, it should conduct training and testing automatically through use of user-specified configuration files, without users' manual execution.

**3. Large-Scale Benchmark.** The tool should allow for large-scale benchmarking, including training and testing of different vision models on various datasets when the number of classes varies.

## A.2  BENCHMARK USAGE GUIDELINE

Users should prepare the dataset detailed in 3.1 in the "CUSTOM" format based on the MMPreTrain (Contributors, 2023) documentation. The tool is designed for both practitioners and researchers.

**Practitioner:** Users select the target $N_{CL}$ and then execute the FC-Sim on the custom dataset. FC-Sim calculates the image similarity score which can be used to index a narrow range of potential target models, given the deployment accuracy requirement (e.g. accuracy). We provide the index table covering ten models on ten datasets for the *Few-Class Regime*.

**Researcher:** Users specify the configurations for FC-Full, FC-Sub and FC-Sim, as well as a list of $N_{CL}$. In the next step, users can execute the scripts running on all benchmarks on the custom dataset. The results for all benchmarks are then used for further analysis.

## A.3  EXTENDED RELATED WORK

**Few-Shot Learning.** There has been a large body of research on Few-Short Learning (FSL) (Song et al., 2023; Wang et al., 2020; Hu et al., 2022; Sung et al., 2018). However, the fundamental research questions differ from ours in the *Few-Class Regime*. The FSL framework aims to address the problem of **data scarsity** with the goal for a model to leverage the representations from very few samples (or none, in the case of Zero-Shot Learning), or prior knowledge that can **generalize** effectively to other tasks or domains.

**Self-Supervised Learning.** To leverage the knowledge from unlabeled data, Self-Supervised Learning (SSL) has emerged as an effective learning framework to learn general vision features (Jaiswal et al., 2020; Jing & Tian, 2020). This includes techniques such as Contrastive Learning, applied to single modalities (Chen et al., 2020a; Chen & He, 2021; He et al., 2020; Chen et al., 2020b; 2021) or multiple modalities (Radford et al., 2021), as well as mask-and-reconstruct methods (He et al., 2022), among others.

In contrast to the advancements of the aforementioned learning frameworks, *Few-Class Arena* focuses on the research problem of selecting the most **efficient** model with **minimal** features (e.g. model parameters, model multiplies) needed for the target application deployment.

## A.4  FULL MODELS ON IMAGENET

In practice, ImageNet serves as a common benchmark for vision neural networks. We list the details of ten pre-trained models from MMPreTrain (Contributors, 2023) in terms of Top-1 Accuracy and scale ($\#Params$) in Fig. 6.

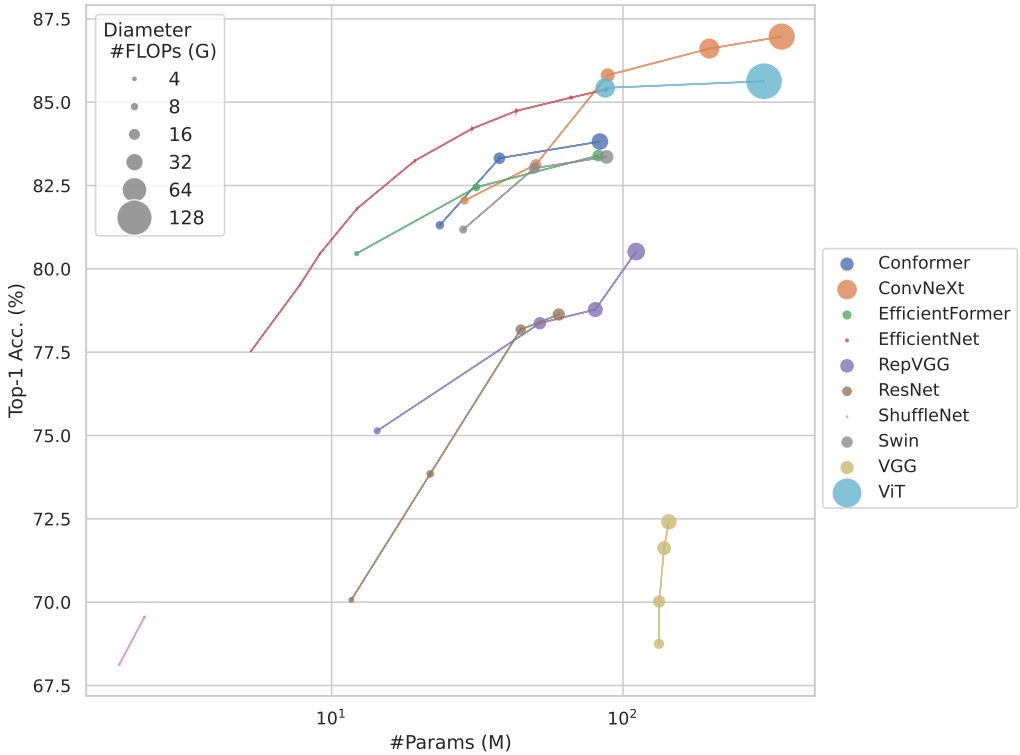

Figure 6: Top-1 Accuracy (%) vs. number of parameters and FLOPs (G) (size of circle) on ImageNet.

| Model | Ref. | Model | Ref. |
|---|---|---|---|
| Conformer | (Peng et al., 2021) | ConvNeXt | (Liu et al., 2022b) |
| EfficientFormer | (Li et al., 2022b) | EfficientNet | (Tan & Le, 2019b) |
| RepVGG | (Ding et al., 2021) | ResNet | (He et al., 2016) |
| ShuffleNet | (Zhang et al., 2017) | Swin | (Liu et al., 2021) |
| VGG | (Simonyan & Zisserman, 2014a) | ViT | (Dosovitskiy et al., 2020) |

Table 3: Full models pre-trained on ImageNet.

## A.5 EXTENDED MANY-CLASS FULL DATASET TRAINED BENCHMARK RESULTS

A complete ranking of ten models in ten datasets is depicted in Fig. 7. Observe that the ten models' rankings differ dramatically among ten different datasets where each line changes from ImageNet1K (IN1K) to other datasets. This poses some questions whether rankings in existing benchmarks can be a reliable indicator for a practitioner to select an efficient neural network, especially when the deployed environment changes from application to application. A major variable in this process is the reduced number of classes from benchmark datasets to deployed environments in the *Few-Class Regime*. As such, our tool is developed to facilitate research in the *Few-Class Regime*.

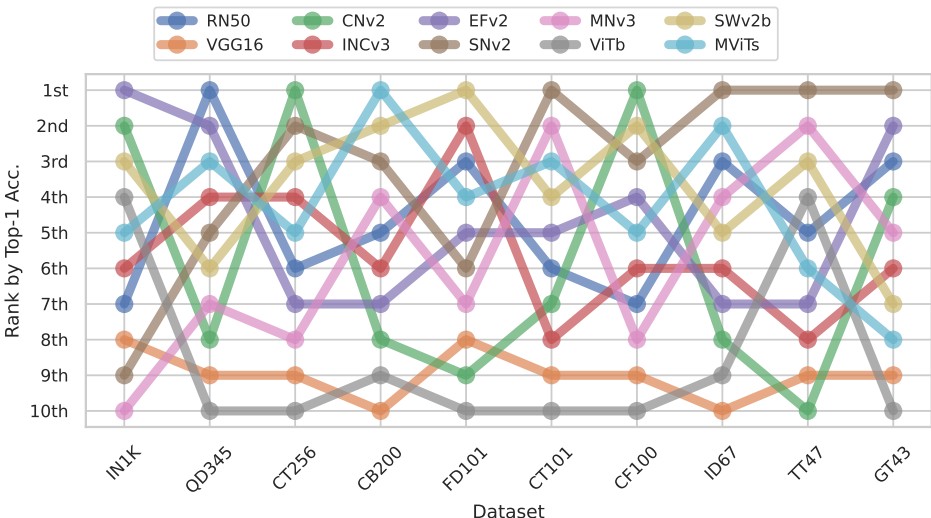

Figure 7: Extended Details of Fig. 2 (b) in the main paper. Full Ranking of ten models across ten datasets by Top-1 acc.

## A.6 DATASETS

Dataset information is presented in Table 4.

| Dataset Name | Abbrev. | Ref. | Homepage | Path in FCA |
|---|---|---|---|---|
| Caltech 101 | CT101 | (Li et al., 2022a) | data.caltech.edu/records/mzrjq-6wc02 | tools/ncls/datasets/caltech101.py |
| Caltech 256 | CT256 | (Griffin et al., 2022) | data.caltech.edu/records/nyy15-4j048 | tools/ncls/datasets/caltech256.py |
| CIFAR-100 | CF100 | (Krizhevsky et al., 2009) | cs.toronto.edu/ kriz/cifar.html

github.com/knjcode/cifar2png | tools/ncls/datasets/cifar100.py |
| Caltech-UCSD Birds-200-2011 | CB200 | (Wah et al., 2011) | vision.caltech.edu/visipedia/CUB-200-2011.html

data.caltech.edu/records/65de6-vp158/files/CUB_200_2011.tgz | tools/ncls/datasets/cub200.py |
| Food 101 | FD101 | (Bossard et al., 2014) | vision.ee.ethz.ch/datasets_extra/food-101/

huggingface.co/datasets/food101 | tools/ncls/datasets/food101.py |
| German Traffic Sign | GT43 | (Stallkamp et al., 2012) | benchmark.ini.rub.de/ | tools/ncls/datasets/gtsrb43.py |
| ImageNet | IN1K | (Deng et al., 2009) | image-net.org/challenges/LSVRC/2012/index.php | * |
| Indoor Scene Recognition | ID67 | (Quattoni & Torralba, 2009) | web.mit.edu/torralba/www/indoor.html | tools/ncls/datasets/indoor67.py |
| Quickdraw | QD345 | (Ha & Eck, 2017) | github.com/googlecreativelab/quickdraw-dataset

tensorflow.org/datasets/community_catalog/huggingface/quickdraw | tools/ncls/datasets/quickdraw345.py |
| Describable Textures Dataset | TT47 | (Cimpoi et al., 2014) | robots.ox.ac.uk/ vgg/data/dtd/index.html | tools/ncls/datasets/textures47.py |

Table 4: Dataset information. * Note that ImageNet dataset format is used as the reference for other datasets. Therefore, the Path in FCA is not required for ImageNet.

**License.** We have searched available online resources and list the license of each dataset in Table 5. For licenses not found in the datasets or websites denoted as "*", we assume they are non-commercial research use only.

| Dataset | License | Dataset | License |
|---------|---------|---------|---------|
| CT101 | CC BY 4.0 | CT256 | CC BY 4.0 |
| CF100 | MIT | CB200 | CC BY 4.0 |
| FD101 | CC BY-SA 4.0 | GT43 | GPLv2 |
| IN1K | * | ID67 | * |
| QD345 | CC BY 4.0 | TT47 | * |

Table 5: Licenses of ten datasets.

**Train/val splits.** The dataset format follows the convention of ImageNet:

```
imagenet1k/
    meta
        train.txt
        val.txt
    train
        <IMAGE_ID>.jpeg
        ...
    val
         <IMAGE_ID>.jpeg
         ...
```

where a .txt file stores a pair of image id and and class number in each row in the following format:

```
<IMAGE_ID>.jpeg <CLASS_NUM>
```

We follow the same train/val splits when the original dataset has already provided. If the dataset does not have explicit splits, we first assign image IDs to all images, starting from 0. We then select $4/5$ of the images as the training set and place the rest in the validation set. Specifially, if an image's ID satisfies the condition $ID \% 5 == 0$, it is moved to the validation set; otherwise, it is assigned as a training sample.

## A.7 MODEL TRAINING DETAILS

Model training details are presented in Table 6.

| Model | Abbrev. | Ref. | Optimizer | LR | Weight Decay | Other Params |
|---|---|---|---|---|---|---|
| ResNet50 | RN50 | (He et al., 2016) | SGD | 0.1 | 1e-4 | momentum=0.9 |
| VGG16 | VGG16 | (Simonyan & Zisserman, 2014b) | SGD | 0.01 | 1e-4 | momentum=0.9 |
| ConvNeXt V2 Base | CNv2 | (Woo et al., 2023) | AdamW | 2.5e-3 | 0.05 | eps=1e-8 betas=(0.9, 0.999) |
| Inception V3 | INCv3 | (Szegedy et al., 2016) | SGD | 0.1 | 1e-4 | momentum=0.9 |
| EfficientNet V2 Medium | EFv2 | (Tan & L., 2021) | SGD | 4e-3 | 0.1 | momentum=0.9 clip_grad: max_norm=5.0 |
| ShuffleNet V2 | SNv2 | (Ma et al., 2018) | SGD | 0.5 | 0.9 | momentum=4e-5 |
| MobileNet V3 Small | MNv3 | (Howard et al., 2019) | RMSprop | 6.4e-4 | 1e-5 | alpha=0.9 momentum=0.9 eps=0.0316 |
| Vision Transformer Base | ViTb | (Dosovitskiy et al., 2020) | AdamW | 3e-3 | 0.3 | - |
| Swin Transformer V2 Base | SWv2b | (Liu et al., 2022a) | AdamW | 1e-4 | 0.05 | eps=1e-8 betas=(0.9, 0.999) |
| MobileViT Small | MViTs | (Mehta & Rastegari) | SGD | 0.1 | 1e-4 | momentum=0.9 |

Table 6: Model Training detials. LR: Learning rate. SGD: Stochastic gradient descent. AdamW: Adam with weight decay. RMSprop: Root mean square propagation.

## A.8 Extended Few-Class Full Dataset Trained Benchmark (FC-Full) Results

We present the details of FC-Full results for each experiment model, including ResNet50, VGG16, ConNeXt V2 Base, Inception V3, EfficientNet V2 Medium, ShuffleNet V2, MobileNet V3 Small, ViT Base, Swin Transformer V2 Base, MobileViT Small in Fig. 8-17, respectively.

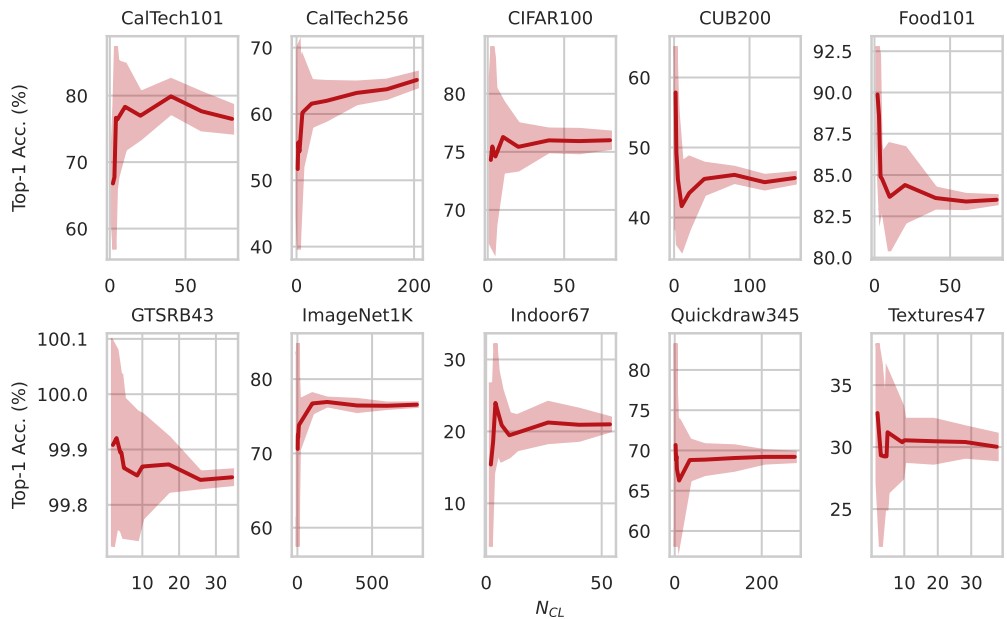

Figure 8: FC-Full Top-1 Accuracy (%) for ResNet50.

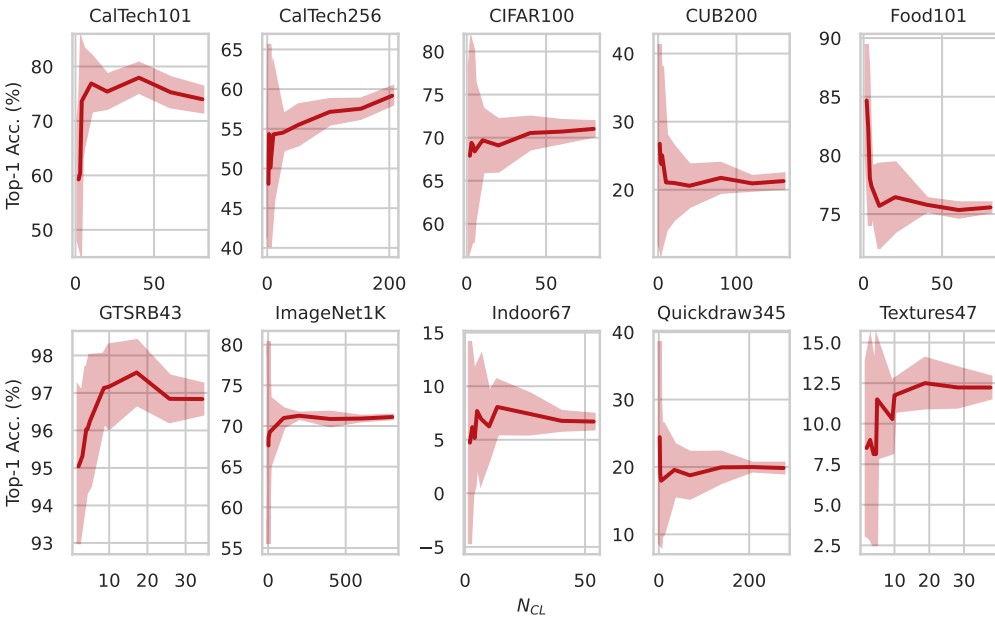

Figure 9: FC-Full Top-1 Accuracy (%) for VGG16.

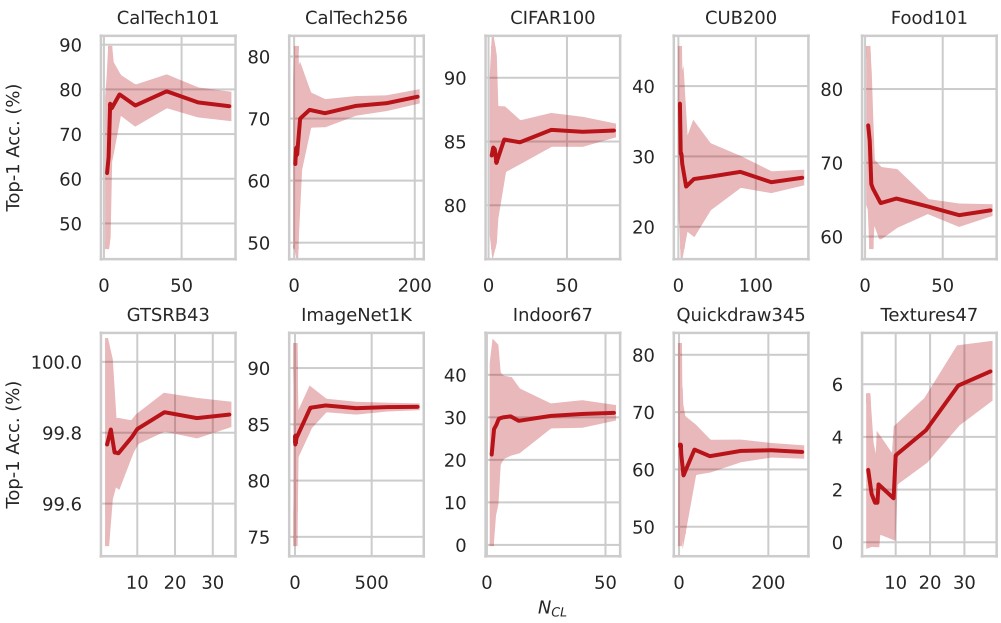

Figure 10: FC-Full Top-1 Accuracy (%) for ConNeXt V2 Base.

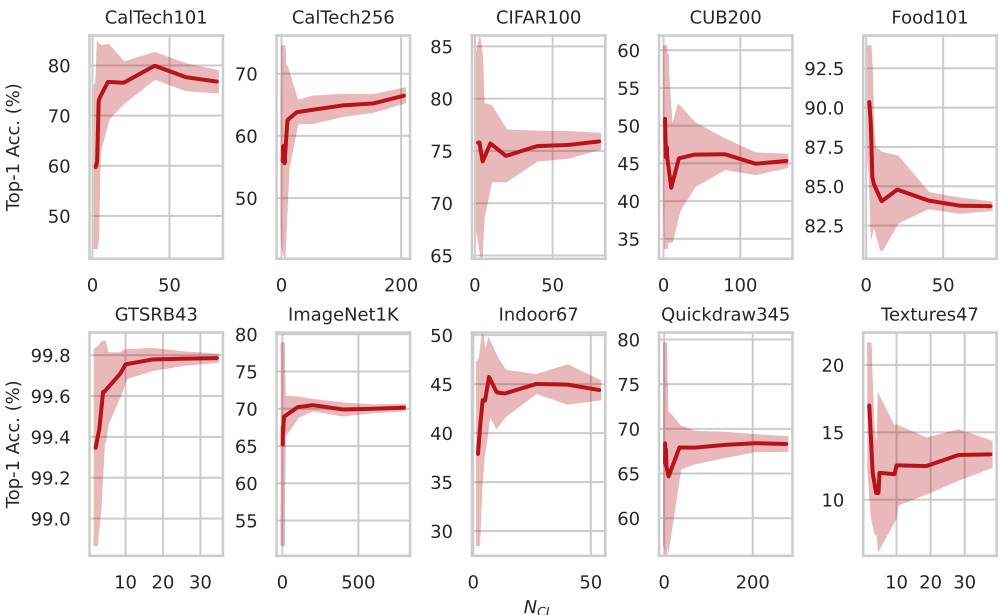

Figure 11: FC-Full Top-1 Accuracy (%) for Inception V3.

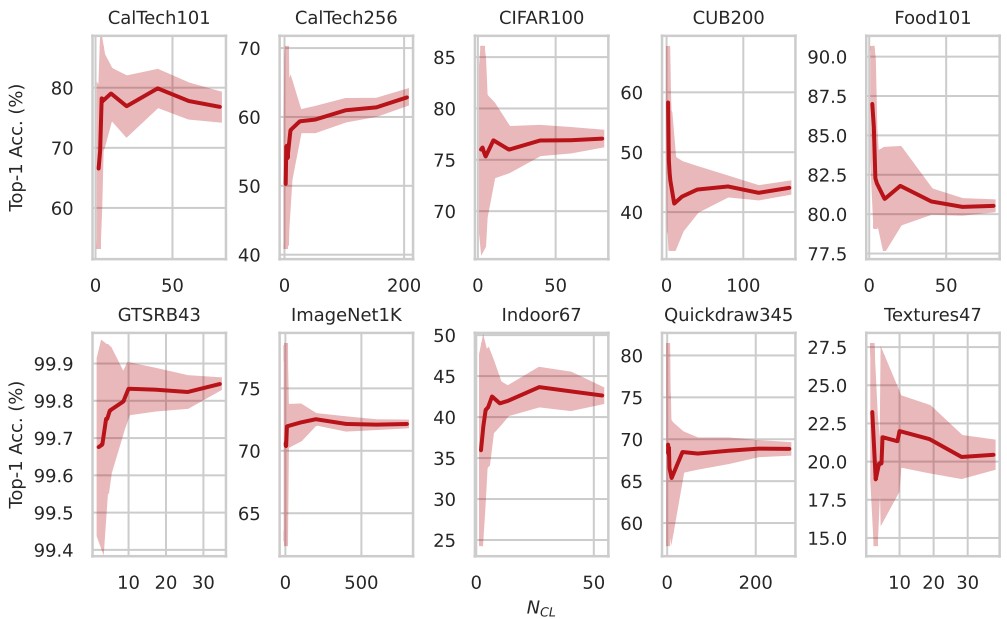

Figure 12: FC-Full Top-1 Accuracy (%) for EfficientNet V2 Medium.

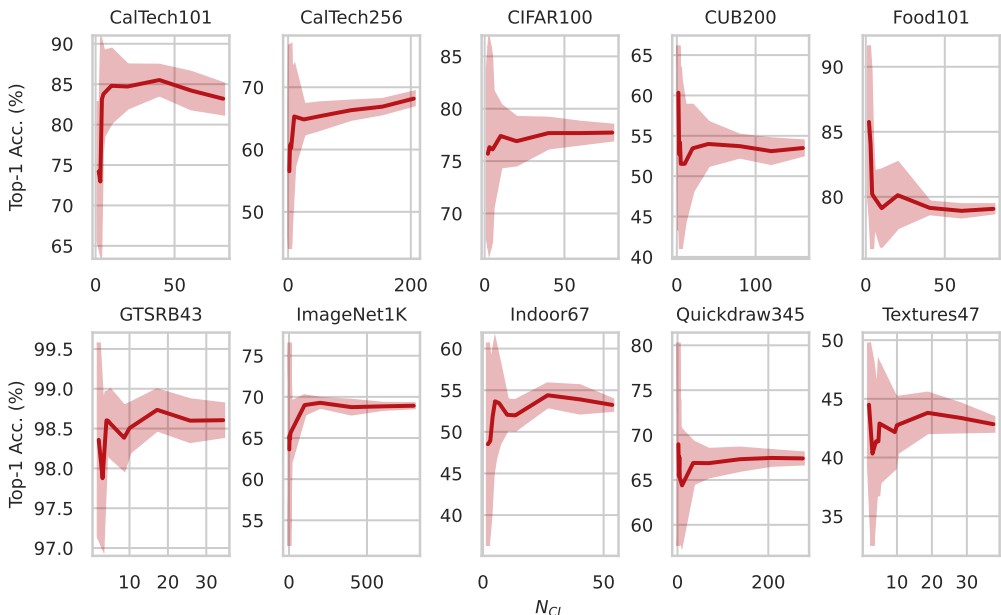

Figure 13: FC-Full Top-1 Accuracy (%) for ShuffleNet V2.

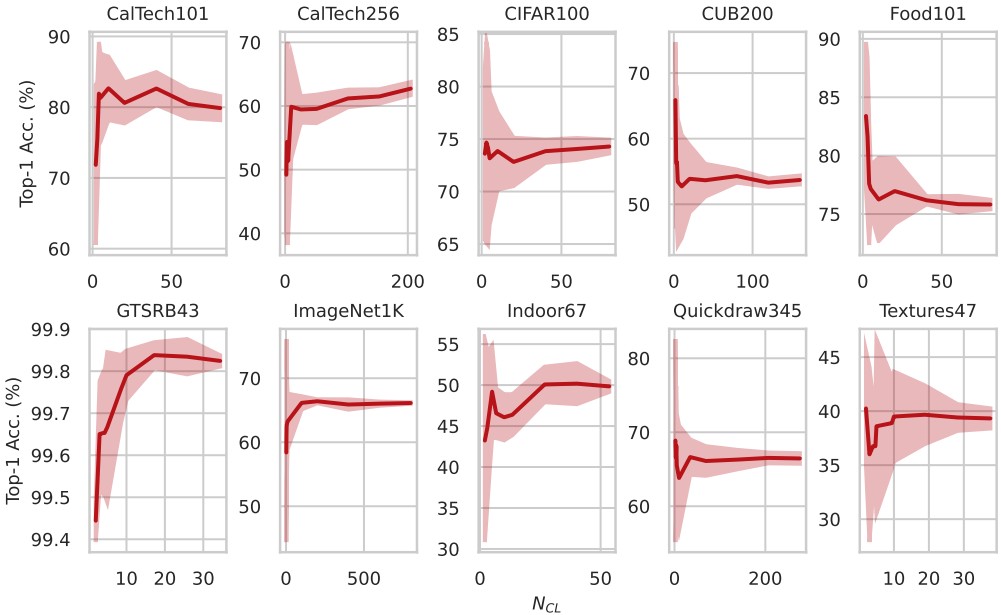

Figure 14: FC-Full Top-1 Accuracy (%) for MobileNet V3 Small.

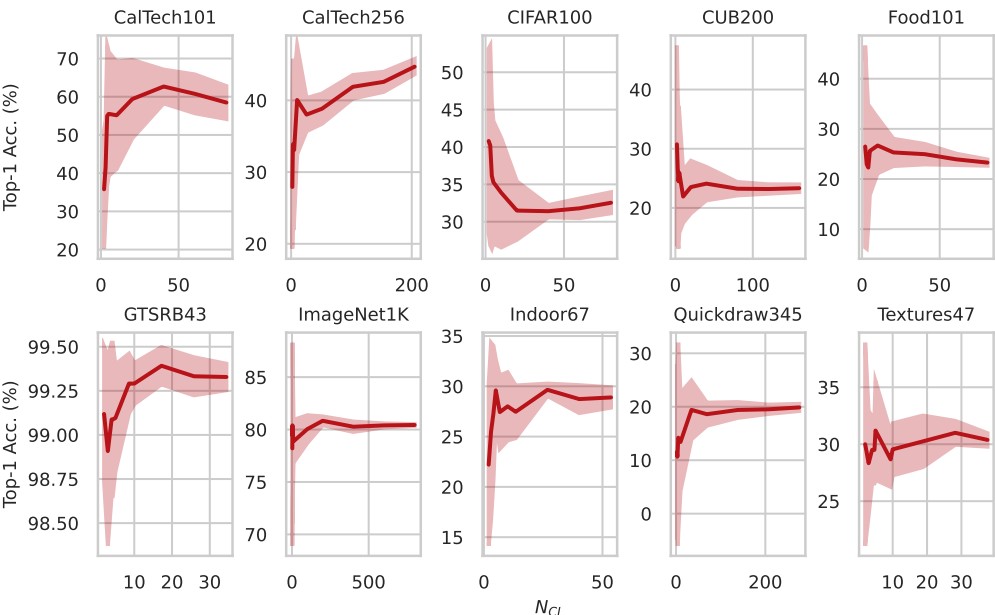

Figure 15: FC-Full Top-1 Accuracy (%) for ViT Base.

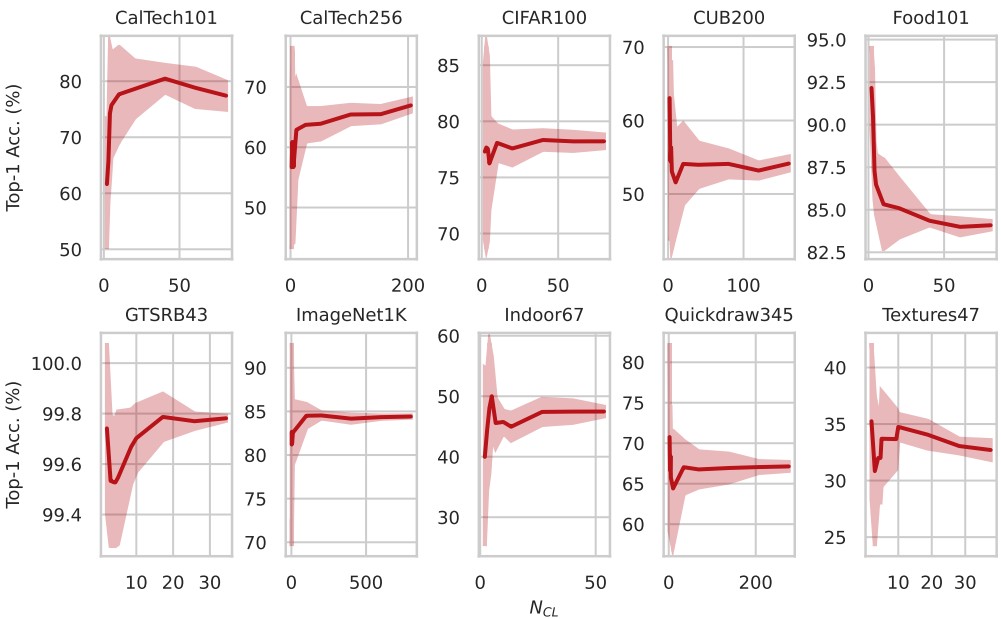

Figure 16: FC-Full Top-1 Accuracy (%) for Swin V2 Base.

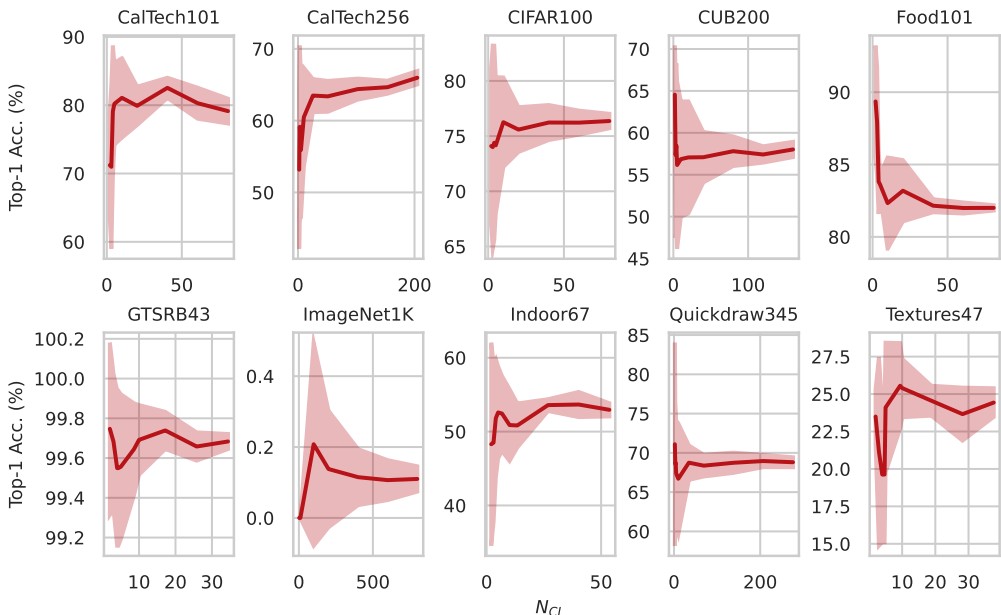

Figure 17: FC-Full Top-1 Accuracy (%) for MobileNetViT Small.

A.9    EXTENDED FEW-CLASS SIMILARITY BENCHMARK (FC-SIM) DETAILS

We present the extended mathematical details of Section 3.5 Few-Class Similarity Benchmark (FC-Sim) in the main paper.

The basic similarity formulation is adopted from (Cao et al., 2023). Notations are defined as follows:

**Dataset** $D$: a set of image instances in a dataset.

**Class** $C$: a set of image instances in a class and $|C|$ is the number of instances within class $C$.

**Class Label** $L$: a set of class labels in a dataset and $|L|$ is the number of classes in a dataset.

**Feature** $Z_i$: visual feature of an image and $i$ is the instance index.

**Class Pair** $P^{(D)}$: a set of distinct class pairs in a dataset $D$; $|P^{(D)}|$ is the total number of distinct class pairs.

**Intra-Class Image Pair** $P^{(C)}$: a set of distinct image pairs in a class $C$; $|P^{(C)}|$ is the total number of distinct image pairs.

**Inter-Class Image Pair** $P^{(C_1,C_2)}$: a set of distinct image pairs in two classes $C_1, C_2$; $|P^{(C_1,C_2)}|$ is the total number of distinct image pairs. Note that this does not include same-class pairs.

**Intra-Class Similarity** $S_\alpha^{(C)}$: a scalar describing the similarity of images within a class by taking the average of all the distinct class pairs in $C$:

$$S_\alpha^{(C)} = \frac{1}{|P^{(C)}|} \sum_{i,j \in C; \ i \neq j} \cos(\mathbf{Z}_i, \mathbf{Z}_j). \tag{5}$$

**Inter-Class Similarity** $S_\beta^{(C_1,C_2)}$: a scalar describing the similarity among images in two different classes $C_1$ and $C_2$:

$$S_\beta^{(C_1,C_2)} = \frac{1}{|P^{(C_1,C_2)}|} \sum_{i \in C_1, j \in C_2} \cos(\mathbf{Z}_i, \mathbf{Z}_j), \tag{6}$$

where $C_1$ and $C_2$ are distinct classes, $i$ and $j$ are the image instance indices in $C_1$ and $C_2$, respectively. $P^{(C_1,C_2)}$ is the set of distinct pairs of images between $C_1$ and $C_2$.

The above equations formulate class-level similarity scores. For dataset-level, Intra-Class Similarity and Inter-Class Similarity of a dataset $D$ are defined as the mean of their similarity scores, respectively:

$$S_\alpha^{(D)} = \frac{1}{|L|} \sum_{l \in L} S_\alpha^{(C_l)} = \frac{1}{|L| \times |P^{(C_l)}|} \sum_{l \in L} \sum_{i,j \in C_l; \ i \neq j} \cos(\mathbf{Z}_i, \mathbf{Z}_j), \tag{7}$$

$$S_\beta^{(D)} = \frac{1}{|P^{(D)}|} \sum_{a,b \in L; a \neq b} S_\beta^{(C_a,C_b)} = \frac{1}{|P^{(D)}| \times |P^{(C_1,C_2)}|} \sum_{a,b \in L; \ a \neq b} \sum_{i \in C_1, j \in C_2} \cos(\mathbf{Z}_i, \mathbf{Z}_j). \tag{8}$$

Averaging these similarities can provide a summary of score in class or dataset levels by a single scalar. However, this simplicity neglects other cluster-related information that can better reveal the underlying dataset difficulty property of a dataset. In particular, the **(1) tightness of a class cluster** and **(2) distance to other classes** of class clusters, are features that characterize the inherent class difficulty, but are not captured by $S_\alpha$ or $S_\beta$ alone.

To compensate the aforementioned drawback, we adopt the Silhouette Score (SS) (also called Silhouette Coefficient in the literature) (Rousseeuw, 1987; Shahapure & Nicholas, 2020):

$$SS(i) = \frac{b(i) - a(i)}{max(a(i), b(i))}, \tag{9}$$

where $SS(i)$ is the Silhouette Score of the data point $i$, $a(i)$ is the average dissimilarity between $i$ and other instances in the same class, and $b(i)$ is the average dissimilarity between $i$ and other data

points in the closest different class. Intuitively, this metric summarizes the quality of clusters by jointly considering the distance between instances of the same class and distinct clusters, normalized by the longest distance of $a(i)$ and $b(i)$. By this definition, we can see that $-1 \leq SS(i) \leq 1$ where $-1$ indicates a dataset is poorly clustered (data points with different classes are scattered around) while $1$ represents a well-clustered dataset.

Euclidean Distance is commonly used to measure two data points' differences; in contrast, we incorporate the inverse of similarity (dissimilarity) as data points' differences into the existing Silhouette Score. Observe that the above Intra-Class Similarity $S_\alpha^{(C)}$ already represents the tightness of the class $(C)$, therefore $a(i)$ can be replaced with the inverse of Intra-Class Similarity $a(i) = -S_\alpha(i)$. For the second term $b(i)$, we adopt the previously defined Inter-Class Similarity $S_\beta^{(C_1, C_2)}$ and introduce a new similarity score as follows:

**Nearest Inter-Class Similarity** $S'^{(C)}_\beta$: a scalar describing the similarity among instances between class $C$ and the closest class of each instance in $C$:

$$S'^{(C)}_\beta = \frac{1}{|P^{(C,\hat{C})}|} \sum_{i \in C, j \in \hat{C}} \cos(\mathbf{Z}_i, \mathbf{Z}_j), \tag{10}$$

where $\hat{C}$ is the nearest class to instance $i$ ($\hat{C} \neq C$). To determine $\hat{C}$, we first iterate over all other classes $C'$ that are different from $C$. For each $C'$, we compute the average similarity between $i$ and all samples in $C'$. The class $C'$ with the highest average similarity score is then chosen as $\hat{C}$.

Consequently, the dataset-level Nearest Inter-Class Similarity $S'^{(D)}_\beta$ is expressed as:

$$S'^{(D)}_\beta = \frac{1}{|L|} \sum_{l \in L} S'^{(C_l)}_\beta = \frac{1}{|L| \times |P^{(C_l, \hat{C}_l)}|} \sum_{l \in L} \sum_{i \in C_l, j \in \hat{C}_l} \cos(\mathbf{Z}_i, \mathbf{Z}_j). \tag{11}$$

The second term of $SS(i)$ can be written as $b(i) = -S'_\beta(i)$.

Replacing $a(i)$ and $b(i)$ from equation 9 with these similarity terms, we introduce our novel similarity metric:

**Similarity-Based Silhouette Score** $SimSS$:

$$SimSS(i) = \frac{S_\alpha(i) - S'_\beta(i)}{max(S_\alpha(i), S'_\beta(i))}, \quad \text{for instance } i \tag{12}$$

$$SimSS^{(C)} = \frac{1}{|C|} \sum_{i \in C} SimSS(i) = \frac{1}{|C|} \sum_{i \in C} \frac{S_\alpha(i) - S'_\beta(i)}{max(S_\alpha(i), S'_\beta(i))}, \quad \text{for class } C \tag{13}$$

$$\begin{aligned} SimSS^{(D)} &= \frac{1}{|L|} \sum_{l \in L} SimSS^{(C_l)} = \frac{1}{|L| \times |C_l|} \sum_{l \in L, i \in C_l} SimSS(i) \\ &= \frac{1}{|L| \times |C_l|} \sum_{i \in C_l} \frac{S_\alpha(i) - S'_\beta(i)}{max(S_\alpha(i), S'_\beta(i))}, \quad \text{for dataset } D. \end{aligned} \tag{14}$$

A.10 EXTENDED FEW-CLASS SIMILARITY BENCHMARK (FC-SIM) RESULTS

We present the relationship of similarity scores using our proposed SimSS and number of classes $N_{CL}$ in ten datasets. CLIP and DINOv2 are used as similarity base functions of SimSS shown in Fig. 18 (a) and (b), respectively.

Overall, a key observation is that the general trend among all ten datasets unveils the inverse relationship between similarity and the number of classes. Specifically, image similarities, which act as proxy of inverse subset difficulty score increases as the number of classes $N_{CL}$ decreases. This reveals that similarity plays a more important role in the *Few-Class Regime* than for datasets with more classes. Therefore, for real applications with few classes, simply downscaling a model blindly without considering class similarity may yield a model selection with sub-optimal efficiency. We propose, therefore, that image similarity must be taken into consideration for existing scaling laws (Kaplan et al., 2020; Rae et al., 2021; Zhai et al., 2022). To that end, *Few-Class Arena* is developed to facilitate future research in this direction.

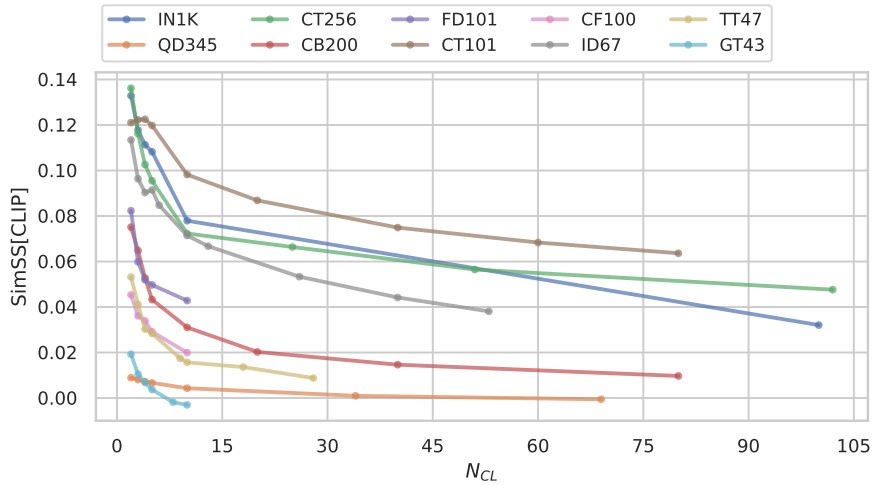

(a) SimSS using CLIP as similarity base function vs $N_{CL}$ curve.

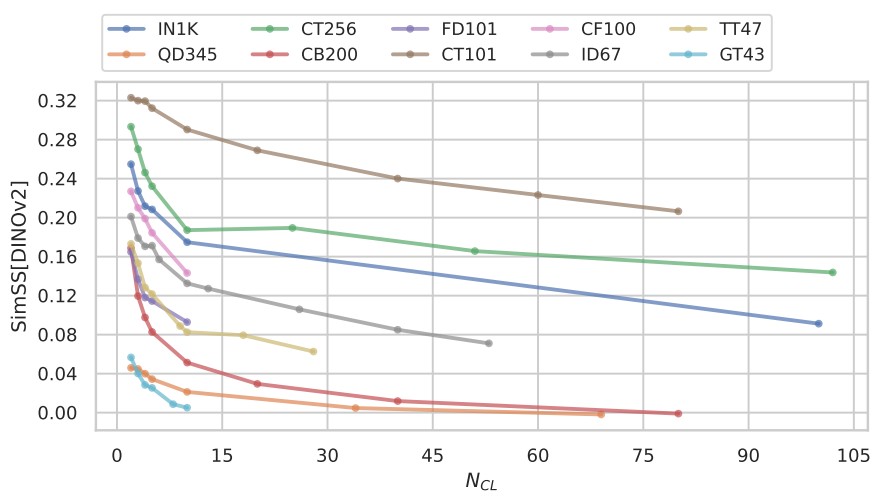

(b) SimSS using DINOv2 as similarity base function vs $N_{CL}$ curve.

Figure 18: Relation of SimSS[CLIP,DINOv2] and $N_{CL}$.

Note that both CLIP and DINOv2 are trained on images from the Internet similar to ImageNet. Therefore to what extent they can capture image similarity in different types is an open research

question. Examples include drawings without textures in the Quickdraw dataset (QD345), textures without shapes in the Describable Textures Dataset (TT47), etc. We mentioned this limitation also in the main manuscript.

**Effect of ResNet Scales on Similarity.** We present the details of FC-Sim results of the ResNet family in different scales in the *Few-Class Regime* of ImageNet1K, specifically ($N_{CL} \in \{2, 3, 4, 5, 10, 100\}$) shown in Fig. 19. In particular, we analyze the relationship between each full and sub-model's Top-1 accuracy and SimSS by Pearson correlation coefficient (PCC) denoted as $r$ in the plots. The ResNet family scales from ResNet18 to ResNet152. We experiment both CLIP (dash line in the 1st and 3rd rows) and DINOv2 (solid line in the 2nd and 4th rows) as similarity base functions.

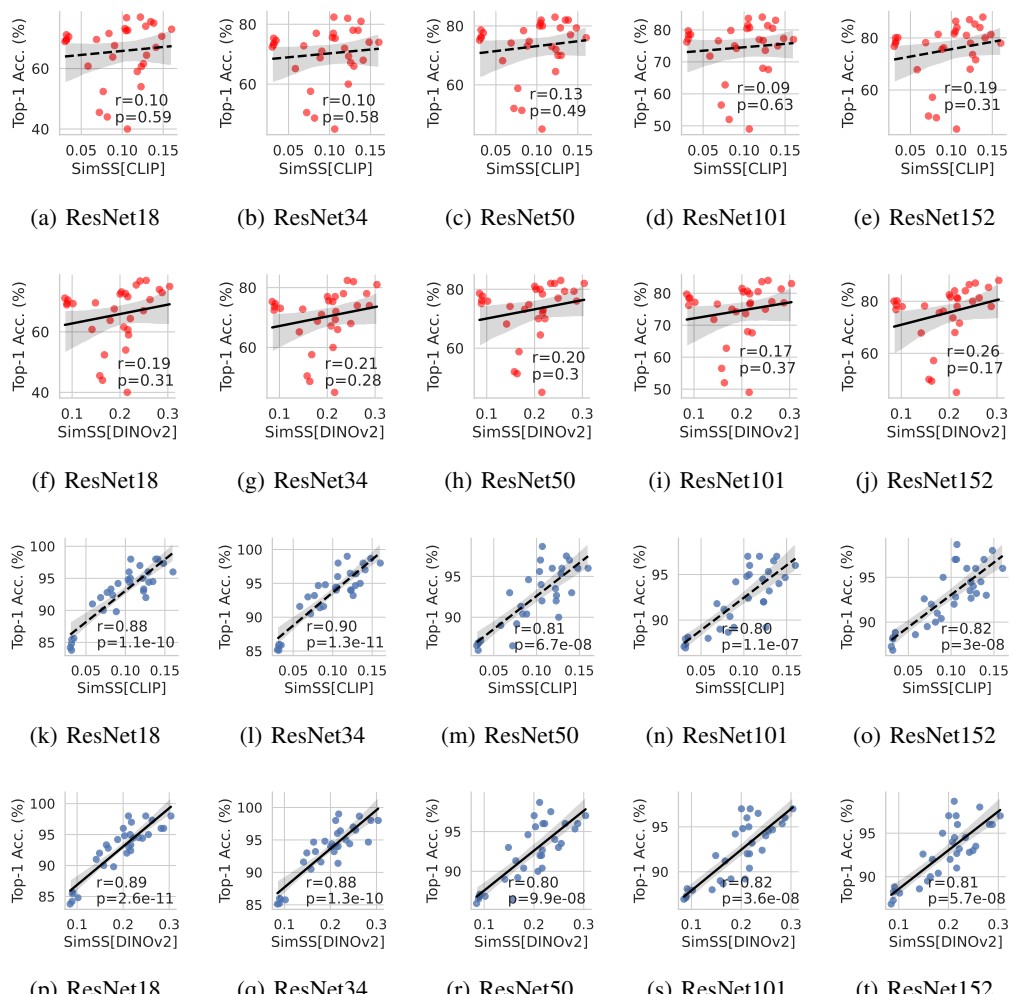

Figure 19: Top-1 Accuracy of different ResNet scales vs SimSS on ImageNet1K. $N_{CL} \in \{2, 3, 4, 5, 10, 100\}$.

In general, ResNet presents high correlation ($r \geq 0.80$) between sub-models' performances and SimSS (blue in the last two rows), compared to full models' performances and SimSS ($r \leq 0.26$, red in the first two rows). This high correlation indicates that SimSS can be used as a reliable tool to estimate upper bound accuracies of ResNet sub-models. Comparing CLIP ($r$ in the 1st row) with DINOv2 ($r$ in the 2nd row) as similarity base functions, observe that PCC is slightly higher for DINOv2 on full models than CLIP, while these differences are subtle for sub-models ($r$ in the 3rd row vs 4th row). Regarding sub-models of ResNet in different scales, the two smallest models'

accuracies (ResNet18 and ResNet34) have higher correlation with SimSS ($r \geq 0.88$), compared to larger models with (ResNet50, ResNet101 and ResNet152) $r \geq 0.80$. We opens a new direction of novel scaling law considering image similarity for efficient models in *Few-Class Regime*.

### A.11 COMPARING VARIOUS ARCHITECTURES

We perform experiments on CNNs (ResNet18, 50) and Vision Transformer-Base (ViT-B). Results are summarized in Table 7. The overall trend of ViT-B exhibits behavior consistent with the observations for ResNet described in Fig. 1.

| MT | $N_{CL}$ | ResNet18 | | ResNet50 | | ViT-B | |
|----|----------|Top-1 Acc.↑|STDEV↓|Top-1 Acc.↑|STDEV↓|Top-1 Acc.↑|STDEV↓|
| F | 1000 | 69.90 | 0 | 76.55 | 0 | 82.37 | 0 |
| F | 10 | 66.68 | 4.372 | 73.76 | 3.675 | 79.52 | 2.704 |
| F | 5 | 65.68 | 9.680 | 72.16 | 9.302 | 79.12 | 8.146 |
| F | 2 | 62.80 | 14.18 | 70.60 | 14.67 | 78.00 | 14.47 |
| S | 10 | 91.88 | 1.640 | 91.48 | 2.265 | 82.16 | 3.508 |
| S | 5 | 93.68 | 1.213 | 92.96 | 3.170 | 89.60 | 2.227 |
| S | 2 | 96.80 | 1.304 | 94.80 | 2.168 | 95.60 | 1.949 |

Table 7: Performance results for different models and configurations on ImageNet1K. $N_{CL} \in \{2, 5, 10, 1000\}$, MT: Model Type, F: Full model, S: Sub-model.

### A.12 EXTENSION TO OBJECT DETECTION AND SEGMENTATION

To assess the generalization of the few-class properties to Object Detection (OD) and Segmentation (Seg), we conducted validation experiments using YOLOv8. The procedure is consistent with the method outlined in Section 3.3 and 3.4. Specifically, for a specific $N_{CL}$ (2 in this example), we randomly sample the $N_{CL}$ classes from the full dataset of COCO, where each consists of five subsets with seed numbers from 0 to 4. We performed experiments with 2, 5, 80. The YOLOv8-nano model was chosen since we focus on efficiency. Image size of 320x320 was used. Model performance was evaluated using the standard metric, mean average precision at an IoU threshold of 0.5 (mAP@50). Table 8 summarizes our results:

| MT | $N_{CL}$ | OD | | Seg | |
|----|----------|mAP@50↑|STDEV↓|mAP@50↑|STDEV↓|
| F | 80 | 0.405 | 0.195 | 0.378 | 0.200 |
| F | 5 | 0.456 | 0.090 | 0.435 | 0.098 |
| FT | 5 | 0.488 | **0.069** | 0.465 | 0.082 |
| S | 5 | **0.503** | **0.069** | **0.474** | **0.084** |
| F | 2 | 0.488 | 0.161 | 0.475 | 0.180 |
| FT | 2 | 0.505 | 0.127 | 0.457 | 0.159 |
| S | 2 | **0.538** | **0.106** | **0.482** | **0.152** |

Table 8: Performance results for different tasks and configurations on COCO. $N_{CL} \in \{2, 5, 80\}$, MT: Model Type, OD: Object Detection, Seg: Segmentation, F: Full model, S: Sub-model, FT: Fine-tuned model. Best scores are highlighted in bold. The gray bar indicates sub-models as the primary focus of this research.

We also include a comparison with fine-tuned pre-trained models (denoted as F). These models are initialized with weights from the pre-trained YOLOv8-nano on the full dataset with 80 classes, and subsequently fine-tuned for a limited number of epochs (five in this example). We conclude that OD and Seg task exhibit similar observations to those we make in Fig. 1.

## A.13 EXPERIMENTS COMPUTE RESOURCES

Experiments are conducted on two internal clusters with the following hardware specifications: (1) 8 NVIDIA RTX A5000 GPUs (24GB), an AMD EPYC 7513 32-Core Processor, and 882GB of RAM; and (2) 8 NVIDIA TITAN Xp GPUs (12GB), an Intel(R) Xeon(R) CPU E5-2650 v4 @2.20GHz, and 126GB of RAM. When GPUs in two clusters are fully utilized, training ten models in nine datasets takes two weeks; obtaining a single experiment result for FC-Full usually takes less than one minute since it only involves inference without training; getting one FC-Sub experiment result takes approximately two days on average depending on the size of subset and model, which includes both training and testing; computing the SimSS in the *Few-Class Regime* for ten datasets takes around three weeks. Additional experiments are conducted on a gaming desktop with the following hardware specifications: 2 NVIDIA RTX 3090 Ti GPUs (24GB), an AMD Ryzen 5 3600 6-Core Processor, and 78.5GB of RAM.

