# OpenReview forum: "Few-Class Arena: A Benchmark for Efficient Selection of Vision Models and Dataset Difficulty Measurement"
_ICLR.cc/2025/Conference — ICLR 2025 Poster_

### Official Review · Reviewer_o3dH · 2024-11-03

**Soundness:** 3
**Presentation:** 3
**Contribution:** 3
**Rating:** 6
**Confidence:** 3

**Summary:**

The paper introduces a benchmark designed to evaluate and select efficient image classification models in scenarios with a limited number of classes. This setting, common in real-world applications (e.g., 2-10 classes), contrasts with widely used benchmarks like ImageNet and COCO, which involve hundreds or thousands of classes. The paper presents FCA as a tool to help researchers and practitioners efficiently select models for few-class tasks. The paper coins the term ``few-class regime'' and presents some interesting non-intuitive insights regarding the performance of models that are pre-trained with many class datasets and then applied in few-class settings.
In addition, they introduce a dataset difficulty metric by inverting image similarity measured via CLIP and DINOv2 features.

**Strengths:**

1. The necessity of a few-class benchmark is well-motivated by a strong finding that models pre-trained on many-class datasets perform worse than expected on few-class datasets which is an issue not addressed in literature thus far.
2. The paper presents some interesting insights that contradict expected model behavior based on intuition.
3. The authors provide a ready to use code base integrated with other frameworks and libraries ensuring usability.

**Weaknesses:**

1. The scope of the study only covers image classification, while it is in principle also applicable to dense prediction tasks where object classes are present (e.g., object detection, semantic segmentation). It would be insightful if the same findings hold for these tasks. The authors could add a discussion or experiments (if available) for other vision tasks.
2. The experiments cover classification tasks based on supervised (pre-)training. However, there is an increasing trend that classification models are fine-tuning based on self-supervised pre-trained models. This paradigm is not covered in this study, and therefore, the findings are limited to the more traditional fully supervised paradigm. The authors could discuss these aspects or add experimentation with self-supervised pre-trained models (if available).

**Questions:**

1. (Related to W2) The finding that the scaling law w.r.t. model size is violated for submodes is interesting. I would be curious if this only applies to supervised training or also to self-supervised pre-training combined with minimal fine-tuning or linear probing. Did you conduct any experiments into this direction or do you have an intuition on that?
2. You mention that you conducted experiments on different architectures such as ResNet’s and ViT’s. However, the results are presented in an aggregated way. Did you find any significant differences between model architectures? Do the findings of Fig 1 equally apply for both architectures?
3. I agree with your description and ad-hoc interpretation of Fig. 5. However, I am missing a discussion on why we see the low correlation for DCN-Full. Do you have any interpretation of this?


**Minor comments:**

- The correct use of author-year citations would improve readability. I.e.: Author (2024) for in-text citations and (Author, 2024) elsewise.

---

> ### Author Response · Authors · 2024-11-26
> **W1: Generalizing to other vision tasks (e.g. object detection and segmentation)**
>
> We thank Reviewer o3dH for the constructive feedback on __recognizing the motivation__, __acknowledging the significant finding that has not been previously overlooked in the literature__ as well as __appreciating the utility of our benchmark tool__. The comments have been instrumental in refining our work. We would like to address each concern separately and apologize for the delay in our response due to our commitment to conducting additional experiments.
>
> __W1: Generalizing to other vision tasks (e.g. object detection and segmentation)__
>
> __Ans:__ We agree on the review that this study on few-class models should be extended to other vision tasks as mentioned in line 115 in our original submission. As the first work and benchmark tool to assist in application and research in the few-class regime, we intend to __remain our focus solely on image classification (IC) in this work__ with object detection (OD) and segmentation (Seg) as future work. This strategic decision aligns with the progressive advancements in the computer vision community, which often begins with IC (such as the ResNet [1], EfficientNet [2], ViT [3], MMPretrain [4]) before moving to more complex tasks of OD or S, such as the R-CNN series [5-6], EfficientDet [7], ViT for OD [8], MMOD[9] etc.
>
> To __assess the generalization of the few-class properties to OD and Seg__, we conducted additional __validation__ experiments using YOLOv8. The procedure is consistent with the method outlined in the Figure 1 caption.
> Specifically, for a specific $N_{CL}$ (2 in this example), we randomly sample the (2) classes from the full dataset of COCO, where each $N_{CL}$ consists of five subsets with seed numbers from 0 to 4. We performed experiments with  $N_{CL} =  $\{2, 5, 80\}. The YOLOv8-nano model was chosen since we focus on efficiency. Image size of 320x320 was used. Model performance was evaluated using the standard metric, mean average precision at an IoU threshold of 0.5 (__mAP@50__). The tables below summarize our results:
>
> ---
>
> __Table of Average Score↑__
> | Model | $N_{CL}$ | OD mAP@50 | Seg mAP@50 |
> |----------|----------|----------|----------|
> | YOLOv8-nano F | 2 | 	0.488 |		0.475 |
> | __YOLOv8-nano S__ | 2 | 	__0.538 (0.150+)__ |		__0.482 (0.007+)__ |
> | YOLOv8-nano F | 5 | 	0.456 |		0.435 |
> | __YOLOv8-nano S__ | 5 | 	__0.503 (0.047+)__ |		__0.474 (0.039+)__ |
> | YOLOv8-nano F | 80 | 	0.405 |		0.378 |
>
> ---
>
> __Table of Standard Deviation↓__
> | Model | $N_{CL}$ | OD mAP@50 | Seg mAP@50 |
> |----------|----------|----------|----------|
> | YOLOv8-nano F | 2 | 	0.161 |		0.180 |
> | __YOLOv8-nano S__ | 2 | 	__0.106 (0.055-)__ |		__0.152 (0.028-)__ |
> | YOLOv8-nano F | 5 | 	0.090 |		0.098 |
> | __YOLOv8-nano S__ | 5 | 	__0.069 (0.021-)__ |		__0.084 (0.014-)__ |
> | YOLOv8-nano F | 80 | 	0.195 |		0.200 |
>
> ---
>
> where each row summarizes the results of 5 models for a given $N_{CL}$. F: Full model; S: Sub-model. Since there is no "few class" for the full dataset ($N_{CL}=80$), the corresponding model consists only of the full model (using the downloaded pre-trained weights). The arrow signs shows the direction of a better value, e.g., ↑ means higher is better.
>
> Overall, these results in OD and S align with the main findings: _Sub-models attain higher upper-bound
> accuracy than full models_, as indicated by the scores highlighted with the (+) sign in Table of Average Scores↑; _The range of accuracy widens for full models at few-classes, which increases the uncertainty of a practitioner selecting a model for few classes. In contrast, sub-models narrow the range_, as shown in Table of Standard Deviation↓.
>
> We will append these results and discussions in the draft shortly.
>
> [1] He et al. "Identity mappings in deep residual networks." ECCV, 2016.
>
> [2] Tan et al. "Efficientnet: Rethinking model scaling for convolutional neural networks." PMLR, 2019.
>
> [3] Dosovitskiy et al. "An image is worth 16x16 words: Transformers for image recognition at scale." arXiv:2010.11929 (2020).
>
> [4] MMPreTrain Contributors. Openmmlab’s pre-training toolbox and benchmark. https://github.com/open-mmlab/mmpretrain, 2023.
>
> [5] He et al. "Mask r-cnn." ICCV. 2017.
>
> [6] Ren et al. "Faster R-CNN: Towards real-time object detection with region proposal networks." TPAMI 2016.
>
> [7] Tan et al. "Efficientdet: Scalable and efficient object detection."CVPR 2020.
>
> [8] Li et al. "Exploring plain vision transformer backbones for object detection." ECCV 2022.
>
> [9] Chen et al. MMDetection: Open mmlab detection toolbox and benchmark. arXiv preprint arXiv:1906.07155, 2019.

---

> ### Author Response · Authors · 2024-11-26
> **Author-year citations**
>
> Thank you for the suggestion of the correct use of author-year citations. We have updated the draft accordingly.
>
> For instance,
>
> "_Typical examples include 1000 classes in ImageNet Deng et al. (2009) for image classification, and 80 object categories in COCO Lin et al. (2014) for object detection. Previous benchmarks also extend vision to multimodal research such as image-text Lee et al. (2024); Le et al. (2024); Laurenc¸on et al. (2024); Bitton et al. (2022)._"
>
> has been corrected as
>
> "_Typical examples include 1000 classes in ImageNet (Deng et al., 2009) for image classification, and 80 object categories in COCO (Lin et al., 2014) for object detection. Previous benchmarks also extend vision to multimodal research such as image-text (Lee et al., 2024; Le et al., 2024; Laurenc ̧on et al., 2024; Bitton et al., 2022)._"

---

> ### Comment · Reviewer_o3dH · 2024-11-26
>
> Thank you for sharing the additional experiments w.r.t. W1 which indeed align with your initial findings for image classification.
>
> I understand that additional experiments are time consuming and not everything can be covered within a rebuttal period.
> However, I would be curious about your thoughts and/or intuition on W2, Q1, Q2, Q3 as well.

---

> > ### Author Response · Authors · 2024-11-26
> > **Making progress**
> >
> > Thank you for your response! I am summarizing the results and will address other questions shortly. Please stay tuned. Appreciate your patience.

---

> > ### Author Response · Authors · 2024-11-30
> > **All Responses Ready**
> >
> > Dear Reviewer o3dH, thank you for taking your time to review our paper. We have provided responses to address all weaknesses and questions you raised. It would be appreciated if you may reconsider these responses. If any more questions arise, we are happy to provide additional clarity.
> >
> > Thank you again for your valuable feedback!

---

> > > ### Comment · Reviewer_o3dH · 2024-12-03
> > >
> > > Thank you for clarifying my questions. I also appreciate the effort of adding additional experiments.
> > >
> > > There are still an open questions regarding the utility / practical usage of your approach (as also pointed out by reviewers trsf and 2XhF):
> > >
> > > In your reply to to review 2XhF you mention that your benchmark is useful when my practical domain "shares a subset of the object categories with the large pre-trained model, but ground truth labels are unavailable".
> > > I agree that when a practitioner wants to use a pre-trained model off-the-shelf for a use case with a specific number of sub-classes, they can refer to your full model (F) benchmark.
> > >
> > > When would a practitioner use the sub-model (S) benchmark? Do I understand correctly that this is to be used in a different scenario when labels and resources to train a model are available?
> > >
> > > Also, could you confirm that I understood the following aspects correctly, as I did not find them explicitly stated in the manuscript:
> > >
> > > a) Sub-models are trained from scratch rather than using the full model weights as a starting point?
> > >
> > > b) For both (S) and fine-tuning (FT) you train the full model not just the last linear layer?
> > >
> > > Assuming my assumptions a) and b) are true, I would like to add the following question:
> > > When looking at accuracies of (S) and (FT), the differences are small, e.g. Table 2 (reported without errors) and Table 8. I would argue that (S) and (FT) are roughly equal and their accuracies are highly correlated. Taking into account that (FT) is computationally cheaper than (S), I would make an argument that (FT) is the more useful few-class metric in practice. What are your thoughts on that? Am I missing a certain property that is unique for (S)?

---

> > > > ### Author Response · Authors · 2024-12-04
> > > >
> > > > We would like to thank Reviewer o3dH03 for taking your time in reviewing our comments, as well as responses to other reviewers. The comments are thoughtful and helpful in improving our work.
> > > >
> > > > Before addressing each concern in detail, we would like to draw the reviewers' attention to the broader perspective of our benchmark. This framework may require a shift in mindset for both researchers and practitioners as they consider the central research question: "what is the simplest baseline model capable of meeting performance criteria within this Few-Class Regime?" (Line 47).
> > > >
> > > > __Q4__: "_When would a practitioner use the sub-model (S) benchmark? Do I understand correctly that this is to be used in a different scenario when labels and resources to train a model are available?_"
> > > >
> > > > __Ans__: Yes, especially when the pre-trained models are difficult to obtain from public resources. FC-Sub, the sub-model (S) benchmark, would be useful in the following use cases of (1) Out-of-distribution (OOD) and (2) Bias Propagation (BP).
> > > >
> > > > (1) (OOD) is the opposite scenario of the previously mentioned practical domain that "shares a subset of the object categories with the large pre-trained model". For example, unusual objects like spaceships, stars or items in medical images, stretches (assuming the full models are unavailable) may not benefit from fine-tuning from datasets like ImageNet.
> > > >
> > > > (2) In some use cases that rely heavily on shape information like classifying sketches, CNNs pre-trained on ImageNet may be biased towards texture [1] (although ViT can help mitigate the texture-bias problem).
> > > >
> > > > In such cases, it would be preferable to train a model from scratch in FC-Sub. Note that our benchmark has already included the Quickdraw345 and Textures47 datasets.
> > > >
> > > > "_Also, could you confirm that I understood the following aspects correctly, as I did not find them explicitly stated in the manuscript:_"
> > > >
> > > > __Q5.a__ "_Sub-models are trained from scratch rather than using the full model weights as a starting point?_"
> > > >
> > > > __Ans__: Yes.
> > > >
> > > > __Q6.b__ "_For both (S) and fine-tuning (FT) you train the full model not just the last linear layer?_"
> > > >
> > > > __Ans__: Yes. Due to time constraints, we did not attempt freezing the backbone and training only the last linear layers. However, the limited number of training epochs during fine-tuning should not significantly alter the visual backbones.
> > > >
> > > > __Q7.a__ "_Assuming my assumptions a) and b) are true, I would like to add the following question: When looking at accuracies of (S) and (FT), the differences are small, e.g. Table 2 (reported without errors) and Table 8._"
> > > >
> > > > __Ans__: Yes. Mentioned on Line 285 in :"Then Few-Class Arena generates bash scripts for model training on each
> > > > subset." in Section 3.4. We will further clarify sub-models by explicitly mentioning "training from stretch" in the final version.
> > > >
> > > > __Q7.b__ "_I would argue that (S) and (FT) are roughly equal and their accuracies are highly correlated._"
> > > >
> > > > __Ans__: Yes, based on the results.
> > > >
> > > > __Q7.c__ "_Taking into account that (FT) is computationally cheaper than (S), I would make an argument that (FT) is the more useful few-class metric in practice. What are your thoughts on that? Am I missing a certain property that is unique for (S)?_"
> > > >
> > > > __Ans__: While I agree that for a __single run of training__, FT is computationally cheaper than S as typically a smaller number of layers (such as the last fully-connected layers while freezing the visual backbone, or additional layers from the backbone can be selectively updated) are being trained for only a few epochs for FT. It will still take __many runs of training__ in FT in order to find the most efficient models (Line 49). The propose SimSS metric in FC-Sim is a __one-time compute on the dataset__ to index that model.
> > > >
> > > > (7.c.1) One of the goals of our benchmark is __Generality__ stated in A.1 GOALS. We aim at making it general for all use cases. Therefore we focus on models trained from scratch in FC-Sub, covering the cases of OOD and BP.
> > > >
> > > > (7.c.2) For researchers, we want the models to fully adapt to the sub-classes without potential extraneous from full datasets (Line 461) for further analysis.
> > > >
> > > > Both (7.c.1) and (7.c.2) make us prioritize sub-models for analysis in our current work.
> > > >
> > > > Given the amount of questions regarding fine-tuning (FT), we have added an option of (FT) for users in our benchmark code: https://github.com/fewclassarena/fca (Search the keyword "Fine-Tuning sub-models").
> > > >
> > > > [1] Geirhos et al. "ImageNet-trained CNNs are biased towards texture; increasing shape bias improves accuracy and robustness." In International Conference on Learning Representations 2019.

---

> ### Author Response · Authors · 2024-11-28
> **W2 & Q1: Discussion on fine-tuned models by pre-trained self-supervised models**
>
> __W2__: "*The experiments cover classification tasks based on supervised (pre-)training. However, there is an increasing trend that classification models are fine-tuning based on self-supervised pre-trained models. This paradigm is not covered in this study, and therefore, the findings are limited to the more traditional fully supervised paradigm. The authors could discuss these aspects or add experimentation with self-supervised pre-trained models (if available).*"
>
> __Q1__: "*(Related to W2) The finding that the scaling law w.r.t. model size is violated for submodes is interesting. I would be curious if this only applies to supervised training or also to self-supervised pre-training combined with minimal fine-tuning or linear probing. Did you conduct any experiments into this direction or do you have an intuition on that?*"
>
> __Ans__: We appreciate the suggestion of pre-trained models by Self-Supervised Learning (SSL) methods, which can potentially bring new insights into the Few-Class Regime study.
>
> We would like to first highlight the __research goal differences__: in the __Few-Class Regime__, our goal is to identify the __smallest baseline__ model that achieves the desired performance accuracy while utilizing the __minimal set of features__ necessary for the target __few-class applications__. This direction is fundamentally different from the SSL framework that aim to learn __general representations__ by leveraging the knowledge from large amounts of __unlabeled data__, typically via a a pretext task. This can further enable models adapted to a target application from a large well pre-trained SSL model. However, we hypothesize that such a (SSL) pre-trained model may have included __extraneous__ features that are not required in the target application, which can contradict with the aforementioned "efficiency" goal in the __Few-Class Regime__ (discussed on Line 384 in Section 4.2 RESULTS ON FC-SUB). To investigate the Few-Class problems, we strategically select sub-models as they are only trained on the target few-classes. We apologize for the lack of clarity regarding their relations and differences, and have addressed this issue in the revised draft. Please refer to Line 47 and Line 845 in A.3 EXTENDED RELATED WORK within the Appendix for a detailed explanation. The new addition is highlighted in blue.
>
> We have included additional study on comparing with fine-tuned (SSL) models in Table 2 in 4.4 COMPARISON WITH FINE-TUNED MODELS in the revision. Table 2 shows the Top-1 Accuracies for different configurations on CIFAR100. $N_{CL} \in$ \{2, 4, 100\}. Best scores are highlighted in bold.
>
> For ViT, we fine-tune a ViT-B model initialized with weights from the CLIP pre-trained backbone. A linear layer is added on top, and the model is trained for 10 epochs. This setup is indicated by the star symbol (*).
>
> ---
>
> __Table 2__
> | Model Type | $N_{CL}$ | ResNet18 | ResNet50 | MobileViT-Small | ViT-B |
> |----------|----------|----------|----------|----------|----------|
> | Full | 100 |  76.11  | 73.71 | 73.83 | 32.54 |
> | Full | 4 | 75.10 |	72.20 | 72.35 | 36.15 |
> | Fine-tuned | 4 | 	87.60 |		__90.55__ | __90.00__ | __91.16__* |
> | Sub | 4 | __90.65__ | 90.15 | 89.45 | 85.40 |
> | Full | 2 | 75.00 |	71.30 | 71.80 | 40.80 |
> | Fine-tuned | 2 | 	87.90 |		93.70 | 90.50 | 95.20* |
> | Sub | 2 | __96.30__ | __95.30__ | __95.50__ | __95.90__ |
>
> ---
>
> We conclude that the fine-tuned models exhibit patterns and trends consistent with the observations presented in Fig. 1. Note our focus of this work is to leverage the proposed difficult measurement method, FC-Sim, to efficiently estimate the achievable model accuracy, thereby assisting in model selection in the Few-Class Regime. Sub-models can offer insights into the minimal visual features required for a specific real-world scenario as they are trained exclusively on the target classes. In contrast, weights pre-trained on large full datasets -- whether through fully supervised or self-supervised manner -- may include extraneous features that are irrelevant to the target classes. We hereby prioritize sub-model study in this work.

---

> > ### Author Response · Authors · 2024-11-28
> > **Q2: Differences between model architectures**
> >
> > __Q2.1__: "_You mention that you conducted experiments on different architectures such as ResNet’s and ViT’s. However, the results are presented in an aggregated way. Did you find any significant differences between model architectures?_"
> >
> > __Ans__: __No clear differences__ between model architectures have been observed so far in terms of the trend from full classes to few classes. Despite some differences in training various architectures, my high-level observations on the ten models on ten datasets indicate that the main factors are the scale of the models and the target dataset difficulty, which motivates us to design and develop the Similarity-Based Silhouette Score to quantify this difficulty.
> >
> > __Q2.2__: "_Do the findings of Fig 1 equally apply for both architectures?_"
> >
> > __Ans__: __Yes.__ We have included the __additional__ experiments on CNNs (ResNet18, 50) and Vision Transformer-Base (ViT-B). Results are summarized in Table 7. The overall trend of __ViT-B__ is __consistent__ with the observations for ResNet described in Fig. 1. We included the results in the A.11 COMPARING VARIOUS ARCHITECTURES section in the revised draft.
> >
> > ---
> > Table 7
> >
> > | Model Type | $N_{CL}$ | ResNet18 | | ResNet50 | | ViT-B |
> > |----------|----------|----------|----------|----------|----------|----------|
> > |  |  |  Top-1Acc.↑  | STDEV↓ | Top-1Acc.↑ | STDEV↓ | Top-1Acc.↑ | STDEV↓ |
> > | Full | 1000 |  69.90 | 0 | 76.55 | 0 | 82.37 | 0 |
> > | Full | 10 | 66.68 | 4.372 | 73.76 | 3.675 | 79.52 | 2.704 |
> > | Full | 5 | 	65.68 | 9.680 | 72.16 | 9.302 | 79.12 | 8.146 |
> > | Full | 2 | 62.80 | 14.18 | 70.60 | 14.67 | 78.00 | 14.47 |
> > | Sub | 10 | 91.88 | 1.640 | 91.48 | 2.265 | 82.16 | 3.508 |
> > | Sub | 5 | 93.68 | 1.213 | 92.96 | 3.170 | 89.60 | 2.227 |
> > | Sub | 2 | 96.80 | 1.304 | 94.80 | 2.168 | 95.60 | 1.949 |
> >
> > Note that ViT-B outperforms both ResNet18 and ResNet50 in the full dataset, achieving a Top-1 accuracy of 82.37%. However, smaller models like ResNet18 or ResNet50 can achieve competitive performance compared to the ViT-B model. While it is well-established in the literature that ViTs generally perform better with large amounts of data, here we aim to revisit the study from a fresh Few-Class perspective, which sets our work apart from prior studies.

---

> > > ### Author Response · Authors · 2024-11-28
> > > **Q3: Discussion on the low correlation for DCN-Full**
> > >
> > > __Q3__: "_I agree with your description and ad-hoc interpretation of Fig. 5. However, I am missing a discussion on why we see the low correlation for DCN-Full. Do you have any interpretation of this?_"
> > >
> > > __Ans__: Apologize for the missing discussion. In a nutshell, the models in DCN-Full, trained by the overall training paradigm in full datasets may have learned __general features__ that could include __extraneous parameters__ not beneficial to some sub classes. This leads to __high variance__ of accuracies within the same sub-classes, meaning that such models' performance does not represent the upper bound of achievable sub-class accuracy. Our proposed __Nearest Inter-Class Similarity__ $S'_{\beta}$ is designed to capture the inherent difficulty of a target (sub)dataset, which, in theory correlates with the highest empirical accuracies (as verified in Fig. 5 (c) and (d)). The fact that the __high variance__ in DCN-Full, which __fails to reflect these sub-class upper limits__, results in the __low correlation__ shown in Fig. 5 (a) and (b).
> > >
> > > We hypothesize that models in DCN-Full (derived from the full models in large full datasets) have learned __general visual representations__ not specific to the target sub-classes. The conventional training paradigm enforces the model to perform __equally well__, if balanced, on all many classes in the full dataset. As a result, some parameters that could potentially be beneficial to a certain sub-class set may need to be __adjusted__, __sacrificing performance on some sub-classes__ for others in order to minimize the overall objective function (typically cross-entropy loss) among all many classes. When such full models are deployed directly to the Few-Class Regime, high variance occurs as the overall cross-entropy loss function does not encourage a model to learn representations specific to each target subset. This leads to the scattered pattern observed in Fig. 5 (a) and (b).
> > >
> > > We would like to bring these under-explored questions to the community, which inspired the development of the Few-Class Arena tool for further study in this area. A summarized discussion is included in Section 4.3 RESULTS ON FC-SIM in blue in the revised version.

---

### Official Review · Reviewer_2XhF · 2024-11-03

**Soundness:** 3
**Presentation:** 3
**Contribution:** 2
**Rating:** 6
**Confidence:** 3

**Summary:**

This paper presents a new benchmark for the “few-class” problem, which  is a classification problem with very few classes. Most of the scientific literature focuses on datasets with many classes while practitioners often encounter the few-class scenario. The benchmark consist of several selected datasets and several settings, such as training on large set of classes and evaluating on a smaller set, and popular vision models are evaluated and compared. Finally, an analysis of what happens in few-class regimes is proposed.

**Strengths:**

- The benchmark is well executed and will be useful for “few-class” adaptation research. There are many models evaluated with many datasets and the analysis is thorough. In particular, the author study in depth the evolution of the performance of models trained on large set of classes compared to specialized models, as a function of the number of classes, and show the importance, and therefore propose a metric for evaluating this adaptation.

- The similarity benchmark is a nice addition. It correlates well with the performance while being easy to evaluate and with a modest cost.

- The presentation and writing are very clear. The figures are very informative.

**Weaknesses:**

- The motivation behind few-class evaluations is not fully convincing. In practice, one will take a large model and fine-tune it (without the classification layer) to a target set of classes, hence obtaining a specialized model. Evaluating the capabilities of a full model on few-class is only interesting when there are too many subsets to consider ? When does that happen in practice, and could you just not use small adaptation layers on top of frozen backbone for each of the subsets ?

- One thing that is missing from the paper is a recommendation for practitioners on which vision model to use for someone interested in the few-class problem. Basically discussing in more details the results from Table 1 and providing comparison between models in Section 4.2 and 4.3. One interesting question is, do models that perform really well on the many classes setup are the same that also perform well on the few-class setup ?

- Some of the findings in the paper are fully expected. The fact that a model specifically trained on the target subset of classes perform better that a larger model trained on a superset is not very surprising or novel.

**Questions:**

- What original research do you expect will use this benchmark and what do you hope it will achieve or unlock ?

---

> ### Author Response · Authors · 2024-11-30
> **W1: Adaptation Layers in Transfer Learning**
>
> We sincerely thank Reviewer 2XhF for appreciating the breadth of our work ("many models with many datasets"), the thoroughness and depth of the trained models, the strong correlation demonstrated between the proposed similarity measure and performance, as well as the clarity and quality of our presentation and writing.
>
> __W1.1__: _"The motivation behind few-class evaluations is not fully convincing. In practice, one will take a large model and fine-tune it (without the classification layer) to a target set of classes, hence obtaining a specialized model. "_
>
> __Ans__: We agree on the commonly used fine-tuning method to adapt to a target sub classes.
>
> We would like to first highlight the __research goal differences__: in the __Few-Class Regime__, our goal is to identify the __smallest baseline__ model that achieves the desired performance accuracy while utilizing the __minimal set of features (MF)__ necessary for the target __few-class applications__. Fine-tuning, however, does __not explicitly__ take MF into consideration. We hypothesize that such a large pre-trained model may have included __extraneous__ features that are not required in the target application, which can contradict with the aforementioned "efficiency" goal in the __Few-Class Regime__ (discussed on Line 384 in Section 4.2 RESULTS ON FC-SUB). Therefore, we argue that full model evaluation is necessary to investigate the Few-Class problems. Please refer to Line 47 and Line 845 in A.3 EXTENDED RELATED WORK within the Appendix for a detailed explanation. The new addition is highlighted in blue.
>
> We have included __additional__ study on comparing with __fine-tuned__ pre-trained models in Table 2 in 4.4 COMPARISON WITH FINE-TUNED MODELS in the revision. We also include the results of transfer learning from a Self-Supervised pre-trained model ViT-B. Table 2 shows the Top-1 Accuracies for different configurations on CIFAR100. $N_{CL} \in$ \{2, 4, 100\}. Best scores are highlighted in bold.
>
> For ViT, we fine-tune a ViT-B model initialized with weights from the CLIP pre-trained backbone. A linear layer is added on top, and the model is trained for 10 epochs. This setup is indicated by the star symbol (*).
>
> ---
>
> __Table 2__
> | Model Type | $N_{CL}$ | ResNet18 | ResNet50 | MobileViT-Small | ViT-B |
> |----------|----------|----------|----------|----------|----------|
> | Full | 100 |  76.11  | 73.71 | 73.83 | 32.54 |
> | Full | 4 | 75.10 |	72.20 | 72.35 | 36.15 |
> | Fine-tuned | 4 | 	87.60 |		__90.55__ | __90.00__ | __91.16__* |
> | Sub | 4 | __90.65__ | 90.15 | 89.45 | 85.40 |
> | Full | 2 | 75.00 |	71.30 | 71.80 | 40.80 |
> | Fine-tuned | 2 | 	87.90 |		93.70 | 90.50 | 95.20* |
> | Sub | 2 | __96.30__ | __95.30__ | __95.50__ | __95.90__ |
>
> ---
>
> We conclude that the fine-tuned models exhibit patterns and trends consistent with the observations presented in Fig. 1. Note our focus of this work is to leverage the proposed difficult measurement method, FC-Sim, to efficiently estimate the achievable model accuracy, thereby assisting in model selection in the Few-Class Regime.
>
> __W1.2__: "_When does that happen in practice, and could you just not use small adaptation layers on top of frozen backbone for each of the subsets ?_"
>
> __Ans__: It happens when the usage environment __shares a subset of the object categories__ with the large pre-trained model, but __ground truth labels are unavailable__ in the target environment, which is very common when users would like to simply apply off-the-shelf models whose weights are pre-trained from large many-class datasets. Popular examples include deploying pre-trained YOLO models directly for industrial use cases, such as detecting general objects like pedestrians and vehicles. (For details on extending to object detection, please see Section A.12 in the revision and our Response to W1: Generalizing to other vision tasks (e.g., object detection and segmentation) for Reviewer o3dH. In this work, however, we focus specifically on image classification.)
>
> When a large model encounters novel classes in the deployment that were not included in its training dataset, ground truth labels are typically required in order to train new adaptation layers for classification. If we don't use any adaptation layers trained by the novel class ground truth labels, the model will simply output the class label with the highest confidence, but the correctness of this prediction cannot be guaranteed."

---

> ### Author Response · Authors · 2024-11-30
> **W2: Recommendation for practitioners**
>
> __W2.1__: "_One thing that is missing from the paper is a recommendation for practitioners on which vision model to use for someone interested in the few-class problem._"
>
> __Ans__: We have added A.2 BENCHMARK USAGE GUIDELINE in the revised draft. We would like to distinguish between the use cases of practitioners (P) and researchers (R): Ps are primarily interested in selecting the optimal, model while Rs would like to study many models in the Few-Class Regime. For P, users can compute the proposed SimSS score to identify and select the model that best satisfies their accuracy and hardware constraints. For R, we assume the main interest is the study of model comparison (full and sub-models in FC-Full and FC-Sub), and accessing difficulty measurements in FC-Sim. Our benchmark tool features in configuration files that cover various scenarios by allowing specifications such as $N_{CL}$, seed numbers and so forth. It provides streamlined interfaces, sparing users from managing such tedious implementations for users to conduct large-scale experiments, detailed in Section 3 FEW-CLASS ARENA (FCA).
>
> __W2.2__: "_Basically discussing in more details the results from Table 1 and providing comparison between models in Section 4.2 and 4.3._"
>
> __Ans__: We have already provided the details (in the original draft) of ten models on the ten datasets from Fig. 8 to Fig. 17 in Section A.8. Due to the large amount of result data, we summarized the comparisons in Fig. 3 and 4 in Section 4.2 and 4.3 and discussed the general observations. We encourage a more concrete question if possible.
>
> __W2.3__: "_One interesting question is, do models that perform really well on the many classes setup are the same that also perform well on the few-class setup ?_"
>
> __Ans__: __Not necessarily__. We show that the rankings differ dramatically for different models on various datasets in Fig. 2 (b) and Fig. 7.

---

> ### Author Response · Authors · 2024-11-30
> **W3: Novel findings**
>
> __W3__: "_Some of the findings in the paper are fully expected. The fact that a model specifically trained on the target subset of classes perform better that a larger model trained on a superset is not very surprising or novel._"
>
> __Ans__: Our paper presents in a progressive way: first revisit some well observed findings in prior work (which are expected) "_(a) Sub-models attain higher upper-bound accuracy than full models.", "(b) The range of accuracy widens for full models at few-classes, which increases the uncertainty of a practitioner selecting a model for few classes. In contrast, sub-models narrow the range._" The __scaling law__ is an emerging general law but mainly describing how models __scale up__: "_(c) Full models follow the scaling law [1] in the dimension of model size - larger models (darker red) have higher accuracy from many to few classes_." Then we present our __novel findings__ of __scaling down__ in the __Few-Class Regime__: "_(d) Surprisingly, the scaling law is violated for sub-models in the Few-Class Regime where larger models do not necessarily perform better than smaller ones._" described from Line 94 - 102 and Fig. 1. The role of __image similarity__ (measured by SimSS as a proxy of dataset difficulty) is __more pronounced__ in the __Few-Class Regime__ than in the Many-Class Regime, as demonstrated across ten datasets with a wide range of $N_{CL}$ in Fig. 18 in Section A.10. Notably, this type of dataset difficulty measurement has not been considered in existing scaling laws, as mentioned in Line 1467.
>
> [1] Jared et al. Scaling laws forneural language preprintarXiv:2001.08361,2020.

---

> ### Author Response · Authors · 2024-11-30
> **Q: Original research to unlock**
>
> __Q__: "_What original research do you expect will use this benchmark and what do you hope it will achieve or unlock ?_"
>
> __Ans__: By using this tool, we aim to explore key research questions in the Few-Class Regime: What is the simplest baseline model capable of meeting performance criteria within this context? What is an effective dataset difficulty measurement method to assist in model selection? How does the scaling law behave in this setting?
>
> When scaling up to a larger dataset, the scaling Law might guide us in scaling up models in terms of data size, model size etc. However, we show that image similarity (as a proxy of dataset difficulty) plays a more and more important role as $N_{CL}$ decreases to the Few-Class Regime, as demonstrated across all datasets in Fig. 18. We argue that simply downscaling a model blindly without considering class similarity may yield a model selection with sub-optimal efficiency. Therefore, our benchmark tool can assist in the next (down) scaling law that takes image similarity (as a way to measure dataset difficulty) into consideration for model selection.
>
> These issues have been discussed from Line 1463 - 1471. We have introduced a new argument on Line 47 in the revised draft.

---

### Official Review · Reviewer_6NMw · 2024-11-04

**Soundness:** 2
**Presentation:** 2
**Contribution:** 2
**Rating:** 6
**Confidence:** 4

**Summary:**

The paper proposes a benchmark tool called Few-Class Arena to benchmark models on different datasets with smaller number of classes (e.g. < 10) and propose a similarity metric called SimSS to measure dataset difficulty measure. They show that ResNet family models trained on full ImageNet 1K classes show reduced performance when tested only for few ImageNet classes (< 10 classes). On the other hand, the same models when trained on smaller number of ImageNet classes from scratch show higher performance on these classes when compared to models trained on all classes of ImageNet 1k. They show that the proposed SimSS metric can serve as a proxy to estimate the upper bound accuracy of model performance on few-class datasets.

**Strengths:**

•	Addressed an important problem by proposing the benchmarking tool.

•	The tool is designed to be user friendly and allows to run wide range of experiments by setting few hyper-parameters. The tool allows to benchmark on custom models and datasets.

•	Provided a behavioral understanding between models trained on large number of classes vs smaller number of classes in the few-class regime.

•	The proposed similarity metric shown to be linearly corelated with the model performance on small number of classes. Such proxy helps to save computation cost and time of conducting various experiments.

**Weaknesses:**

Despite focusing on an interesting problem setting, the analysis shown in the paper has limited scope. Authors shown experiments on models evaluated or trained on smaller number of classes, however there are no details discussed on how these few classes have been selected and how semantically close these few classes to each other? Would the analysis presented differ by choosing the those few classes differently?

The aspect of transfer learning has not been discussed. It is a common practice to finetune ImageNet pretrained models like ResNet50 or ViT, or recent foundation models like CLIP and DINOv2 to different downstream tasks that include adapting or finetuning them on few classes. The analysis presented in the paper is missing this exploration. Is it better to train the models from scratch on the few classes or finetuning these models work better for few classes? Does SimSS score also align on the finetuned models?

To compute SimSS, a score called Nearest Inter-Class Similarity requires a nearest class (C_hat) to the target class (C), it is not clear how this C_hat is acquired.

Overall, I appreciate the motive and tool for benchmarking few-class regime, however the analysis presented in the paper is incomplete, and I suggest authors to extend their analysis.

**Questions:**

1. For FC-Full, when N_{CL} decreases, how to make sure that model predicts only few classes? Are the logits of those few classes are selected to get the prediction and discard logits of all other classes?

2. If a user has a custom dataset with few classes and want to find a model that works better on this custom dataset, it would be helpful to have an explanation on how this benchmark can assist the user in this case.

---------------------------------------------------------------------------------------
Final review: I appreciate authors for their detailed responses to the comments from all the reviewers. Authors comprehensive responses, clarifications, and additional experiments have addressed the key weaknesses. I now vote towards borderline acceptance.

---

> ### Author Response · Authors · 2024-11-30
> **Actionable suggestions**
>
> Dear Reviewer 6NMw, we really appreciate your valuable suggestions. The feedback of "_suggest authors to extend their analysis_" is somewhat vague for us, we would greatly appreciate any actionable suggestions for experiments to help extend our analysis and contribute to improving our work. Thank you!

---

> ### Author Response · Authors · 2024-12-01
> **W1: Few classes selection and semantic closeness**
>
> We thank Reviewer 6NMw for recognizing the utility of our benchmark, the behavioral analysis of models, and the advantages of the proposed similarity metric in correlating with model performance. We would address each concern separately.
>
> __W1__: "_...there are no details discussed on how these few classes have been selected and how semantically close these few classes to each other?_"
>
> __Ans__: __Few classes are randomly sampled by seed numbers from 0 - 4 by default.__ The details were specified on __Line 82__: "_Each model is tested on 5 subsets whose_ $N_{CL}$ _classes are randomly sampled from the original 1000 classes._", __Line 187__: "_Few-Class Arena generates the specific model and dataset configuration files for each subset, where subset classes are randomly extracted from the full set of classes, as specified by the seed number._" and __Line 412__: "_For reproducible results, we use seed numbers from 0 to 4 to generate 5 subsets for one_ $N_{CL}$ _by default._" For extension to Object Detection and Segmentation, these are detailed from __Line 1610 to 1613__: "_The procedure is consistent with the method outlined in Section3.3 and 3.4. Specifically, for a specific_ $N_{CL}$ (2 in this example), we randomly sample the_ $N_{CL}$ _classes from the full dataset of COCO, where each consists of five subsets with seed numbers from 0 to 4. We performed experiments with 2, 5, 80._"
>
> The proposed Similarity-Based Silhouette Score (SimSS) is designed to capture the __semantic cluster__ characteristics of few classes, in particular "_the (1) tightness of a class cluster and (2) distance to other classes of class clusters, are features that characterize the inherent class difficulty,_" described on __Line 315__. We kindly recommend that the reviewer examine __Fig. 18__ closely for detailed insights into the semantic closeness across ten datasets, spanning from the Many-Class to the __Few-Class Regime__.

---

> ### Author Response · Authors · 2024-12-01
> **W2: Transfer Learning**
>
> __W2__: "_The aspect of transfer learning has not been discussed. It is a common practice to finetune ImageNet pretrained models like ResNet50 or ViT, or recent foundation models like CLIP and DINOv2 to different downstream tasks that include adapting or finetuning them on few classes. The analysis presented in the paper is missing this exploration. Is it better to train  the models from scratch on the few classes or finetuning these models work better for few classes? Does SimSS score also align on the finetuned models?_"
>
> __Ans__: For transfer learning, we conducted __additional__ experiments during the rebuttal period. Please kindly refer to the response to _W1: Adaptation Layers in Transfer Learning_ by Reviewer 2XhF and _W2 & Q1: Discussion on fine-tuned models by pre-trained self-supervised models_ by Reviewer o3dH. The empirical observation is that the performance of the finetuned models is close to the sub-models, therefore the SimSS scores should align with the finetuned models.

---

> ### Author Response · Authors · 2024-12-01
> **W3: How is C_hat acquired?**
>
> __W3__: "_To compute SimSS, a score called Nearest Inter-Class Similarity requires a nearest class (C_hat) to the target class (C), it is not clear how this C_hat is acquired._"
>
> __Ans__: As described on Line 326, it is "_a scalar describing the similarity among instances between class C and the closest class of each instance in C_".
>
> It is computed simply using a __max()__ function (where a higher similarity score indicates greater closeness) applied to the class candidate list (sim_c_p_ls) in our released code:
>
> max(sim_c_p_ls)
>
> https://github.com/fewclassarena/fca/blob/aa796880953a58f79b243a855d7aad3a221b8587/configs/_base_/sim.py#L220

---

> ### Author Response · Authors · 2024-12-01
> **Q1: Full model predictions in the Few-Class Regime**
>
> __Q1__: "_For FC-Full, when_ $N_{CL}$ _decreases, how to make sure that model predicts only few classes? Are the logits of those few classes are selected to get the prediction and discard logits of all other classes?_"
>
> __Ans__: __No modification of the logits (or any layers).__ The few classes are a subset of the full dataset, we let the full models output the labels (predictions) and compare them against the ground truth label to calculate the Top-1 accuracy.

---

> ### Author Response · Authors · 2024-12-01
> **Q2: Benchmark usage guideline**
>
> __Q2__: "_if a user has a custom dataset with few classes and want to find a model that works better on this custom dataset, it would be helpful to have an explanation on how this benchmark can assist the user in this case._"
>
> __Ans__: We have added A.2 BENCHMARK USAGE GUIDELINE in the revised draft. Please refer to the response to "Q1: Difficulty metric usage guideline" by Reviewer trsf and "W2: Recommendation for practitioners" by Reviewer 2XhF.

---

### Official Review · Reviewer_trsf · 2024-11-05

**Soundness:** 3
**Presentation:** 2
**Contribution:** 2
**Rating:** 5
**Confidence:** 3

**Summary:**

This paper tackles the problem of choosing image classifiers for tasks with only a small number of categories ("Few-Class"). To do so, they introduce a new benchmark, termed "Few Class Arena" (FCA) on which they train and evaluate a range of models on subsets of various full datasets (e.g wit so-called sub-models trained on between 2 and all 1k ImageNet categories). The FCA benchmark is open-sourced with code available on GitHub. The paper provides detailed discussion of how the open-source package can be used for model selection in the few-class setting.

Overall, the authors show that models trained on specific sub-classes (sub-models) are better than models trained on the full dataset and evaluated on the same sub-classes, across model sizes. They further show that there is no single best model architecture for a given dataset, and that training models on different datasets result in different rankings of architecture. The authors also propose a "dataset difficulty" metric which can be computed without training a model, and correlates well with the few-class performance of a model on a dataset.

**Strengths:**

- Tab 1 / Fig 2b shows the results when training different base architectures on different classification datasets from scratch. Interestingly, not all results are correlated with the trend on ImageNet-1K, indicating the optimal architecture choice depends on the dataset.
- The code is open sourced and well documented. It seems that it would be simple for a researcher to reproduce the authors claim with limited effort (though I have not run the code myself).
- It is interesting that sub-models consistently outperform full models on ImageNet. The fact that full models have seen more training datapoints in total may have compensated for fewer classes, which makes the result not totally intuitive.

**Weaknesses:**

My main issue with this paper is in overall utility. The high level goal of the paper is to provide a tool with which practitioners can select a model (dominantly through the lens of model *architecture*) for a few-class classification task. The tool basically allows authors to train a model (with most results presented from scratch) on subsets of a given dataset. However, this does not align with the practical problem to me, where practitioners might take a model pretrained on a large amount of *data* (e.g DINOv2 or CLIP) is finetuned for a given task (note that lightweight variants of these models are also open-source).

Given that this paper is predominantly an empirical examination which proposes a practical open-source library, I feel that the lack of experiments with pretrained models prevents acceptance.

Other issues:

- The citation format makes the main text quite difficult to parse
- L52: Main text does not seem to describe Figure 1 accurately?
- There is no discussion of few-shot literature, which is at least tangentially related to this problem

**Questions:**

* I may have missed something, but I cannot understand exactly how the proposed difficulty metric is intended to be used?
* The proposed difficulty metric seems expensive to compute, with pairwise similarity scores required for large subsets of the data. How does this compare to the cost of conducting a single model training run?

---

> ### Author Response · Authors · 2024-11-26
> **Responses to Weaknesses MO1 & M02**
>
> We sincerely thank Reviewer trsf for __acknowledging the limitations we identified in commonly used many-class datasets__, as well as __our efforts in open sourcing our code with detailed documentation__. We also appreciate Reviewer trsf's __recognition of our new findings in the few-class regime__ that challenge the prevailing assumptions and motivate us to explore further into this direction. This has led to the development of the first benchmark specifically designed for few-class problems.
>
> We would like to emphasize that, in addition to releasing our benchmark tool, we have also conducted extensive large-scale experiments. Our key findings, summarized in the Statement of Contributions encompassing a diverse range of architectures of CNNs and transformers, evaluated on 10 datasets with a total of 1591 training and testing runs.
>
> The original submission includes comprehensive results for pre-trained models, referred to as Full-models, which are summarized in Figure 3. These Full-models, including ResNet50, VGG16, ConNeXt V2 Base, Inception V3, EfficientNet V2 Medium, ShuffleNet V2, MobileNet V3 Small, ViT Base, Swin Transformer V2 Base and MobileViT Small have been evaluated on ten datasets for various number of classes. Detailed results for each model are presented in Figures 8-17 in the Appendix.
>
>
> __MO1: Citation format.__
>
> __Ans__: We have corrected the citation format. Please check the revision.
>
>
> __MO2: L52: Main text does not seem to describe Figure 1 accurately?__
>
> __Ans__: The main text summarizes the key findings in Figure 1. Sorry, I do not understand in what specific aspect the main text does not describe Figure 1. It would be great if you can elaborate the points concretely.

---

> > ### Author Response · Authors · 2024-11-28
> > **MO3: Discussion on Few-Shot Learning**
> >
> > __MO3__: "_There is no discussion of few-shot literature, which is at least tangentially related to this problem_"
> >
> > __Ans__: Thank you for the suggestion of discussion the relation between Few-Shot Learning (FSL) and our work. We would like to clarify that the fundamental research questions in FSL differ from ours in the Few-Class Regime. The FSL framework aims to address the problem of __data scarsity__ with the goal for a model to leverage the representations from very __few samples__ (or none, in the case of Zero-Shot Learning), or prior knowledge that can __generalize__ effectively to other tasks or domains.
> >
> > In stead of proposing a new learning frameworks, our Few-Class Arena focuses on the research problem of selecting the most __efficient__ model with __minimal__ features needed for the target application deployment.
> >
> > We have clarified this on line 47 and provided a discussion on FSL in Section A.3 EXTENDED RELATED WORK, in the revised draft.

---

> > > ### Author Response · Authors · 2024-11-28
> > > **Q1: Difficulty metric usage guideline**
> > >
> > > __Q1__: "_I may have missed something, but I cannot understand exactly how the proposed difficulty metric is intended to be used?_"
> > >
> > > __Ans__: A user only needs to provide the target dataset with class labels, it will compute the difficulty score. Recall our benchmark tool is designed for both practitioners and researchers. A practitioner can follow the guidelines detailed in the README.md file in the GitHub link (https://github.com/fewclassarena/fca) and run the execute command. The returned difficulty score can be further used to index the smaller target range of models. In contrast, a research may have interests in analyzing a set of difficulty scores in various sub-class sets, our tool includes the feature of utilizing configuration files to conduct large-scale experiments. By specifying these configurations, a research can only execute once and wait for all results. We include a high-level guidelines in A.2 BENCHMARK USAGE GUIDELINE. We encourage users to refer to our GitHub link for the detailed usage.

---

> ### Author Response · Authors · 2024-11-28
> **Q2: difficulty metric compute complexity, comparison to a single model training run**
>
> __Q2__: "_The proposed difficulty metric seems expensive to compute, with pairwise similarity scores required for large subsets of the data. How does this compare to the cost of conducting a single model training run?_"
>
> __Ans__: Since the difficulty metric involves pairwise computations, the time complexity is quadratic, which can be a problem when the number of classes is large. However, our target usage is in the Few-Class Regime with few classes. Based on our empirical testing on ten datasets, it takes around a few minutes to obtain results for $N_{CL}<10$. The cost is affordable compared to the hours or days required for model training.
>
> We would like to emphasize our motivation that without such a difficult score, it requires the re-evaluation of published models or even retraining (for many training and testing runs) to find an optimal model in an expensive architectural search space.

---

### Meta-Review · Area_Chair_YtJK · 2024-12-26

**Metareview:**

This work aims to benchmark image classification in the few-class regime as a proxy for performance on real-world tasks of this size. Such a focus is complementary to existing large-scale benchmarks such as ImageNet, COCO, etc. with 10s of classes or 1000 classes. The contributions include experiments over many subsets of classes on ImageNet, evaluations of popular model architectures like ResNets and ViTs, and a metric that incorporates class similarity into a difficulty score. The application of these contributions is to guide model selection, to identify the most efficient architecture for a given set of classes, and provide scaling predictions in this regime, and these applications are shown by this work. The benchmark is released as an open-source project for reproduction and future work.

Strengths: The topic is complementary to the majority of work focusing either on (A) the large-scale regime of many classes and a lot of data or (B) the few-shot regime of scarce data. The experiments are thorough in covering different model architectures, numbers of classes, and learning with and without transfer. The proposed difficulty metric can save experimentation time (6NMw).

Weaknesses: The evaluation of the full-class models is fair but incomplete in not conditioning on the subclasses at all: the full models could predict the argmax of the logits for the classes intersecting with a given subset. There is doubt about the utility of the contributed benchmark and analysis to the community, because transfer learning and prompting are nearly ubiquitous (trsf, 6NMw).

Decision: This is a borderline submission, and reviewers questioned the motivation and the justification of its experimental scope. However, the revision and additional results have proven the use of the benchmark, even though it is has a somewhat narrow scope of minimal/efficient architectures that are sufficient for few-class accuracy, and reviewers were convinced to maintain or improve their ratings. The proposed difficulty metric and how it might guide model and dataset scaling brings another lens to scaling questions that are now faced in research and practice alike. Given these uses, and the full open-source release of the benchmark, the meta-reviewer sides with acceptance. Congratulations!

**Additional Comments On Reviewer Discussion:**

Reviewers are borderline with ratings of marginal accept (6NMw: 6, 2XhF: 6, o3dH: 6) and marginal reject (trsf: 5). Reviewers shared key concerns including the unspecified relationship to few-shot learning, the exclusion of fine-tuning from pre-training in the scope of the benchmark and experiments even though it is a standard practice, and some concerns about the motivation and use of the benchmark. The authors provide a thorough and multi-step response to each review. 1/4 reviewers discusses with the authors (03dH) and 2/4 reviewers (6NMw, 03dH) discuss with the AC following the rebuttal and author discussion phase. During AC discussion acknowledge the rebuttal and additional experiments, and raise (6NMw: 5 to 6) or maintain (o3dH: 6) their  borderline positive rating, but do not champion the paper. In the absence of discussion by other reviewers, the meta-reviewer has closely examined the points of each review and the resulting thread of responses by the authors. The main issue raised by trsf was the lack of experiments with pre-training and fine-tuning, which is addressed by the revision and rebuttal experiments and the new fine-tuning option in the proposed library, and the main issue raised by 2XhF is motivation and use, which is addressed by the application to guiding model selection.

By fixing the serious omission of fine-tuning, and showing an application for which the proposed benchmark can spare total experimentation resources, the revision and rebuttal have addressed the most serious shortcomings as weighed by the meta-reviewer.

---

### Decision · Program_Chairs · 2025-01-22

Accept (Poster)